# Janus: Dual-Server Multi-Round Secure Aggregation with Verifiability for Federated Learning

Lang Pu [1]  Jingjing Gu [1]  Chao Lin [2]  Xinyi Huang [3]

## Abstract

Secure Aggregation (SA) is a cornerstone of Federated Learning (FL), ensuring that user updates remain hidden from servers. The advanced Flamingo (S&P'23) has realized multi-round aggregation and improved efficiency. However, it still faces several key challenges: scalability issues with dynamic user participation, a lack of verifiability for server-side aggregation results, and vulnerability to Model Inconsistency Attacks (MIA) caused by a malicious server distributing inconsistent models. To address these issues, we propose *Janus*, a generic SA scheme based on dual-server architecture. Janus ensures security against up to $n-2$ colluding clients (where $n$ is the total client count), which prevents privacy breaches for non-colluders. Additionally, Janus is model-independent, ensuring applicability across any FL model without specific adaptations. Furthermore, Janus introduces a new cryptographic primitive, Separable Homomorphic Commitment, which enables clients to efficiently verify the correctness of aggregation. Finally, extensive experiments show that Janus not only significantly enhances security but also reduces per-client communication and computation overhead from logarithmic to constant scale, with a tolerable impact on model performance.

## 1. Introduction

Traditional machine learning relies on centralized training, where the entire dataset is stored in a single central location and directly accessible by the server. However, users are generally unwilling to share their raw data, particularly when it involves sensitive information, such as medical records, photographs, or confidential business information. Federated Learning (FL) (McMahan et al., 2017) is a distributed learning framework proposed to protect user privacy by enabling collaborative model training without exposing private data, thus encouraging greater user participation. Unfortunately, it has been demonstrated that an adversary can invert a single model update to reveal sensitive information about the target user's local dataset (Hitaj et al., 2017; Nasr et al., 2019; Zhu et al., 2019).

Secure Aggregation (SA) (Bonawitz et al., 2017) is designed to enhance privacy by preventing the adversary from accessing individual update. SA has been regarded as one of the most robust defenses against gradient inversion and inference attacks (Huang et al., 2021). Most existing SA schemes rely on the double-mask technique, which involves time-consuming secret sharing and key agreement. BBSA (Bell et al., 2020) optimizes the communication graph to reduce time-consuming operations and significantly improve overall efficiency. However, it remains limited to single-round aggregation. In contrast, FL typically requires multiple rounds of aggregation to achieve model convergence. Although the protocol can be executed multiple times to support multi-round aggregation, the setup phase must be re-run in each round to maintain privacy. Consequently, the server must interact with all clients every round, leading to heavy communication overhead and reduced efficiency.

Recently, the state-of-the-art Flamingo (Ma et al., 2023) eliminates the re-setup in each round, which supports multi-round SA based on the BBSA. It also optimizes the communication graph to improve the system's performance by introducing a set of decryptors to handle part of the computation. Although Flamingo demonstrates significant advantages in efficiency and functionality, it also has the following limitations. Firstly, the complicated setup procedures of Flamingo limit its practicality in dynamic environments where users frequently join or leave. Secondly, the server can exploit the Model Inconsistency Attacks (MIA) (Pasquini et al., 2022), which result from a malicious server distributing inconsistent models to infer users' local datasets. Lastly,

---

[1]College of Computer Science and Technology, Nanjing University of Aeronautics and Astronautics, Nanjing, China [2]College of Computer and Cyber Security, Fujian Normal University, Fuzhou, China [3]College of Cyber Security, Jinan University, Guangzhou, China. Correspondence to: Jingjing Gu < gujingjing@nuaa.edu.cn>, Chao Lin < linchao91@fjnu.edu.cn>.

*Proceedings of the 42nd International Conference on Machine Learning*, Vancouver, Canada. PMLR 267, 2025. Copyright 2025 by the author(s).

*Table 1.* Comparison of SA Constructions

| Scheme | Input Privacy | Multi-round | Verifiability | Dynamic | Versatility | NS* | Efficience‡ | MIA |
|---|---|---|---|---|---|---|---|---|
| SecAgg (Bonawitz et al., 2017) | ✓ | ✗ | ✗ | ✗ | ✗ | 1 | ◯ | ✗ |
| BBSA (Bell et al., 2020) | ✓ | ✗ | ✗ | ✗ | ✗ | 1 | ◔ | ✗ |
| VeriFL (Guo et al., 2020) | ✓ | ✗ | ✓ | ✗ | ✗ | 1 | ◑ | ✗ |
| ELSA (Rathee et al., 2023) | ✓ | ✗ | ✗ | ✗ | ✗ | 2 | ◕ | ✗ |
| Flamingo (Ma et al., 2023) | ✓ | ✓ | ✗ | ✗ | ✗ | 2† | ◕ | ✗ |
| Janus | ✓ | ✓ | ✓ | ✓ | ✓ | 2 | ● | ✓ |

✓ Support, ✗ No support. Versatility: A generic construction. ⋆ Number of servers. † The decryptors of this construction can be abstracted to a server. ‡ More black parts in the circle indicate better efficiency, and the theoretical support comes from the computation efficiency analysis in Table 2.

users cannot verify if the server correctly performed the aggregation or omitted data.

MIA stems from the reliance on a single server, which is common due to its simplicity. The single-server architecture inherently has access to the aggregated results, thereby exposing the system to potential MIA (Pasquini et al., 2022). Specifically, the server distributes carefully crafted parameters to non-target users, which can trigger the *dying-ReLU* effect that causes non-target users to generate zero gradients during aggregation. As a result, the aggregated gradient effectively reveals the target user's gradient. This attack affects not only double-mask schemes but also all schemes in which the server has access to the aggregation results. Cryptographic signatures can prevent this by allowing users to verify the consistency of received aggregated models. However, they incur heavy computation and require users to negotiate the consistency of the received information, placing a large burden on the system.

Fortunately, our research indicates that preventing MIA requires limiting the server's access to the final aggregation results. To achieve this, we propose a dual-server architecture: one server handles the collection and aggregation of masked gradients, while the other manages the aggregation of all masks. Our method ensures the privacy of non-colluding users remains uncompromised, even in the presence of collusion between the servers and up to $n-2$ users, where $n$ is the total number of users. The malicious server has access only to the aggregated ciphertext of two uncolluding clients, which prevents the extraction of individual gradients. If a single server were compromised, individual privacy would remain intact due to the secure cryptographic primitives that prevent unauthorized access to private data.

The dual-server assumption is feasible in real-world contexts involving entities with different interests, such as banks and other financial institutions, or hospitals and other healthcare organizations. They are motivated to collaborate for the benefit of users and avoid collusion. Similarly, in the Flamingo scheme, the decryptors can also be regarded as one server, forming a dual-server architecture together with the aggregation server. This approach ensures security while leveraging the practical willingness of institutions to cooperate for SA.

However, the dual-server architecture still faces the same challenge as Flamingo: the inability to efficiently verify the aggregation results. Specifically, a malicious server may perform incorrect collection or aggregation of masked gradients or masks. The server may choose faster but less accurate computations to save resources, which can result in incorrect aggregation results. Since servers are often semi-trusted, they could also deliberately mishandle some gradients or falsify aggregation results, misleading users about the training results (Hahn et al., 2021). Most existing schemes achieve verifiable aggregation through homomorphic hashing and homomorphic signatures (Guo et al., 2020; Xu et al., 2019). However, these time-consuming operations introduce significant overhead to the system, resulting in low efficiency. Moreover, errors in aggregation could arise from malicious client submissions, yet current methods fail to enforce strong client-side commitments.

To address these challenges, we introduce a new cryptographic primitive called separable homomorphic commitment (SHC), which ensures both server-side integrity and client-side data accuracy in the dual-server setting. Specifically, our main contributions are summarized as follows.

- Firstly, Janus is the first generic construction of SA based on dual-server architecture, which is well-suited for multi-round aggregation in FL without time-consuming re-setup. By requiring only the servers' public keys, our design eliminates the overhead associated with heavy communication graphs, such as complete graphs or $k$-regular graphs. Additionally, Janus relies on lightweight components, which significantly improves system efficiency.
- Secondly, the conceptual development of a new cryptographic primitive (SHC) for a dual-server architecture to jointly achieve verifiability and enhance privacy. SHC is a homomorphic commitment scheme that supports the separation of messages and random components. It ensures that aggregation results remain invisible to servers, preventing malicious servers from launching MIA. Additionally, we identify a blueprint

that SHC can be instantiated to provide novel verification methods for aggregation results.

- Lastly, we implemented an instantiation for Janus and evaluated it with similar classical schemes via extensive experiments on different models and datasets. The results show that Janus outperforms in terms of both computation and communication. It reduces per-client overhead from the logarithmic scale of current advanced methods to a constant scale. Table 1 demonstrates that Janus surpasses other state-of-the-art schemes in terms of security, efficiency, and functionality.

## 2. Preliminaries

### 2.1. Traditional Cryptographic Commitments

Commitments (Pedersen, 1991) provide the cryptographic cornerstone for integrity and trust in various secure schemes. It enables participants to commit to values without compromising the underlying confidential information. Typically, a non-interactive secure commitment scheme consists of the following three algorithms.

(1) $\mathsf{CSetup}(1^\lambda) \to pp$. The system initialization algorithm takes as input a security parameter $\lambda$, and outputs the public parameter $pp$ for the commitment scheme.

(2) $\mathsf{Commit}(pp, v, r) \to c$. The commitment generation algorithm takes as input a message $v$ from the message space $\mathcal{M}_{pp}$ and a random number (blinder) $r$ in the randomness space $\mathcal{R}_{pp}$, and outputs the commitment $c$ in the commitment space $\mathcal{C}_{pp}$.

(3) $\mathsf{Reveal}(pp, v, c, r) \to b$. The revealing commitment algorithm takes as input a message $v$, a commitment $c$, and a blinder $r$. If it accepts, then the output $b = 1$; otherwise, $b = 0$.

Normally, a secure commitment scheme must satisfy the following three properties.

- *Completeness.* It ensures that if both the committer and the verifier follow the protocol correctly, the verifier will always accept the decommitment ($\mathsf{Reveal}$).

$$\Pr\left(\begin{array}{c} \mathsf{CSetup}(1^\lambda) \to pp; \\ \mathsf{Commit}(pp, v, r) \to c; \\ \mathsf{Reveal}(pp, v, c, r) = 1 \end{array}\right) = 1. \quad (1)$$

- *Hiding.* During the commitment phase, the verifier cannot infer the committed value from the commitment. It can ensure that the committed value remains confidential until it is revealed. For any $v_1, v_2$ of equal length, and any $r$, the following probability distributions are computationally indistinguishable.

$$\begin{array}{c} \{\mathsf{Commit}(pp, v_1, r) \to c_1\} \stackrel{c}{\approx} \\ \{\mathsf{Commit}(pp, v_2, r) \to c_2\}. \end{array} \quad (2)$$

- *Binding.* After the commitment is made, the committer cannot change the committed value. It can prevent the committer from cheating by ensuring the immutability of the commitment. There exists a negligible function $\mathsf{negl}(\lambda)$ such that for all non-uniform Probabilistic Polynomial Time ($\mathcal{PPT}$) adversaries $\mathcal{A}$,

$$\Pr\left(\begin{array}{c} \mathsf{CSetup}(1^\lambda) \to pp; \\ \mathcal{A}(pp) \to (c, r, v_1, v_2): \\ \mathsf{Reveal}(pp, c, v_1, r_1) = 1 \wedge \\ \mathsf{Reveal}(pp, c, v_2, r_2) = 1 \wedge \\ v_1 \neq v_2 \end{array}\right) \leq \mathsf{negl}(\lambda). \quad (3)$$

### 2.2. Masking-based Secure Aggregation

The One-Time Pad (OTP) is a type of classical encryption that achieves perfect secrecy (Katz & Lindell, 2014). Specifically, a formal OTP scheme usually contains the following main algorithms.

(1) $\mathsf{Masking}(x, k) \to \hat{x}$. The masking algorithm takes as input a secret message $x$ and a private key $k$, and outputs the encryption result $\hat{x}$.

(2) $\mathsf{UnMasking}(\hat{x}, k) \to x$. The unmasking algorithm takes as input an encrypted message $\hat{x}$ and a private key $k$, and outputs the plain message $x$.

SA is designed to prevent centralized servers from accessing individual updates. Due to its simplicity and efficiency, OTP is commonly used in mask-based SA for FL. Users apply OTP-based masking to their updates before uploading to the central server. There are several variants of SA to address different threat models and system requirements. We focus on masking-based aggregation schemes (Bonawitz et al., 2017). Specifically, let $\mathcal{U}$ be a set of users, where each user $u_i \in \mathcal{U}$ holds a private update $x_i$. In masking-based SA, each $u_i$ adds a pair-wise additive mask to get the masked vector $y_i$ as follows.

$$y_i = x_i + \sum_{u_j \in \mathcal{U}: i < j} \mathrm{PRG}(s_{i,j}) - \sum_{u_j \in \mathcal{U}: i > j} \mathrm{PRG}(s_{j,i}), \quad (4)$$

where the pseudorandom generator (PRG) can randomly generate a sequence number based on the random seed $s_{i,j}$. Note that the masks will be removed when all masked input updates $Y_i = \sum_{u_i \in \mathcal{U}} y_i$ are aggregated as follows.

$$Y_i = \sum_{u_i \in \mathcal{U}} (x_i + \sum_{i < j} \mathrm{PRG}(s_{i,j}) - \sum_{i > j} \mathrm{PRG}(s_{j,i})). \quad (5)$$

Obviously, we have $Y_i = \sum_{u_i \in \mathcal{U}} x_i$. To deal with dropped users during the execution, the Shamir secret sharing scheme (Shamir, 1979) is used to share seeds among all users. The Diffie-Hellman (DH) key exchange protocol (Diffie & Hellman, 1976) is used to negotiate the seeds

$s_{i,j}$ for each pair of users $(u_i, u_j) \in \mathcal{U}$. Note that for large-scale FL applications, the above scheme is not effective. For a $n$-user FL system, it takes $\mathcal{O}(n^2)$ communication rounds to run the pairwise DH key exchange protocol.

## 2.3. Model Inconsistency Attacks

At a high level, a Model Inconsistency Attack (MIA) (Pasquini et al., 2022) works by sending a correct model to the target user while distributing crafted ones to non-target users. These crafted model parameters are designed to suppress non-target users' contributions, effectively driving their next-round updates to zero. As a result, the aggregated model primarily reflects the target user's update, breaking the privacy guarantees of SA. Formally, a malicious server $\mathcal{A}$ intends to obtain raw data about the model update of a target user $U_{tar,t}$. It can elaborately distribute constructed parameters $\theta_{i,t}$ to the non-target users $\{\mathcal{U} \setminus U_{tar,t}\}$ and then send normal parameters $\theta_{tar,t}$ to the target user, where $\theta_{i,t} \neq \theta_{tar,t}$ and $\mathcal{U}$ denotes the set of all users. This can trigger the *dying-ReLU* (Lu et al., 2019), where the dead layer cannot generate any gradient. Therefore, the non-target user ends up generating tampered model updates $\Delta_{D_{i,t}}^{\theta_{i,t}}$, where the $D_{i,t}$ is the local data of $U_{i,t}$. Since the parameters of $U_{tar,t}$ are real, it generates a valid update $\Delta_{D_{tar,t}}^{\theta_{tar,t}}$ on its local data $\mathcal{D}_{tar,t}$ in round $t$. These tampered model updates enable $\mathcal{A}$ to obtain the model updates $\Delta_{D_{tar,t}}^{\theta_{tar,t}}$ of $\hat{U}_{tar,t}$ in plaintext. Specifically, the final result of SA (denoted as $\sum$) is as follows,

$$
\begin{aligned}
&\mathcal{A}^{\sum}(\Delta_{D_{1,t}}^{\theta_{1,t}}, ..., \Delta_{D_{i-1,t}}^{\theta_{i-1,t}}, \Delta_{D_{tar,t}}^{\theta_{tar,t}}, \Delta_{D_{i+1,t}}^{\theta_{i+1,t}}, ..., \Delta_{D_{n,t}}^{\theta_{n,t}}) \\
&= \mathcal{A}^{\sum}(0, ..., 0, \Delta_{D_{tar,t}}^{\theta_{tar,t}}, 0, ..., 0) = \Delta_{D_{tar,t}}^{\theta_{tar,t}}.
\end{aligned} \tag{6}
$$

Once $\mathcal{A}$ gets the update $\Delta_{D_{tar,t}}^{\theta_{tar,t}}$, it can get sensitive information about $\mathcal{D}_{tar,t}$ by executing gradient inversion attacks or inference attacks.

## 3. Methods

**Notations.** In this section, we design Janus, a generic privacy-enhanced multi-round SA scheme through a dual-server architecture, where SHC is the core cryptography for verifiability. To facilitate understanding, we first present a new primitive SHC, followed by elaborating on the construction of Janus. Let $\bigodot$ denote the consecutive operation of $\odot$. Specifically, $\bigodot_{i=1}^{n} x_i = x_1 \odot x_2 ... \odot x_n$, where the $\odot$ indicates addition or multiplication depending on the specific scheme. $T$ is the total number of rounds required for the model to converge and $t$ denotes the current round. Let $n$ users participate in the FL training, where users are denoted by $\mathcal{U}_t = \{U_{i,t}, i \in [1, n]\}$. All users negotiate a model architecture and train the model locally on their private data sets $\mathcal{D}_{i,t}$. There are three types of entities in our

system, which are the aggregation server $S_0$, the assistant server $S_1$, and users. We assume that each user $U_{i,t} \in \mathcal{U}_t$ holds a private update $x_{i,t}$ of dimension $m$. For simplicity, we assume that the elements of $x_{i,t}$ and $\sum_{U_{i,t} \in \mathcal{U}_t} x_{i,t}$ are in $\mathbb{Z}_R$ for $R$.

### 3.1. Separable Homomorphic Commitment

**Definition 3.1.** (Separable Homomorphic Commitment, SHC). We define a secure SHC scheme as a cryptographic protocol that enables secure and flexible commitments. It consists of a set of algorithms denoted by a tuple of (Setup, Commit, Se, PCommit, Reveal). The formal syntax of each algorithm is described as follows.

(1) Setup$(1^\lambda) \to pp$. A $\mathcal{PPT}$ initialization algorithm takes as input a security parameter $\lambda$, and outputs a public parameter $pp$.

(2) Commit$(pp, m, r) \to c$. A $\mathcal{PPT}$ commitment algorithm takes as input a public parameter $pp$, a message $m$, and a random number $r$, and outputs a complete commitment $c = (c_m, c_r)$, where $c_m$ is associated with $m$ and $c_r$ is related to the random number (blinder) $r$.

(3) Se$(pp, c, c_r) \to c_m$. A Decisional Polynomial Time ($\mathcal{DPT}$) separation algorithm takes as input a public parameter $pp$, a complete commitment $c$, and a blinder-related part $c_r$, and outputs the message-related commitment $c_m$.

(4) PCommit$(pp, m) \to c_m$. A $\mathcal{DPT}$ commitment algorithm takes as input a public parameter $pp$, a message $m$, and outputs the message-related commitment $c_m$.

(5) Reveal$(pp, c, m, r) \to \{1/0\}$. A $\mathcal{DPT}$ revealing commitment algorithm takes as input the public parameter $pp$, the complete commitment $c$, the message $m$, and the random blinder $r$, and outputs 1 if the $m$ is the valid committed message of $c$ and 0 otherwise.

The SHC can separate the message-related part and compare it with the aggregated results to ensure the correctness of the aggregation. Therefore, two servers independently aggregate the different components of the commitments in the following Janus. In addition to the completeness, binding, and hiding properties of traditional commitment schemes discussed in Section 2, SHC offers the following two additional properties.

- *Separability.* The complete commitment $c$ can be divided into two parts $c = (c_m, c_r)$, where $c_m$ is the part associated with the commitment message $m$ and $c_r$ is related to the random blinder $r$. It can use $c_r$ to extract from the complete commitment $c$ only the part that is relevant to $m$. Taking the classic Pedersen commitment (Pedersen, 1991) as an example, the complete

commitment is $c = h^r g^m$. Given $c_r = h^r, c_m = g^m$, $c_m$ can be computed through $c/c_r$. Furthermore, the $c_m$ can be calculated through $\mathsf{PCommit}(m, pp) = g^m$, where $g$ is a public parameter.

- *Homomorphism.* Homomorphism plays a key role in enabling secure aggregation. Define the space of message and blinder as $\mathcal{M}_c, \mathcal{R}_c$ respectively. For $\forall (m_0, r_0), (m_1, r_1) \in \mathcal{M}_c \times \mathcal{R}_c$, we have

$$\mathsf{Commit}(m_0 + m_1; r_0 + r_1) = \\ \mathsf{Commit}(m_0; r_0) \cdot \mathsf{Commit}(m_1; r_1). \quad (7)$$

SHC's flexible design not only supports SA in FL but also enables seamless adaptation to a wide range of applications that demand both privacy and verifiability. These applications include, but are not limited to, the following scenarios: (1) medical data federation where SHC enables verifiable auditing; (2) dual-server e-voting systems needing verifiable tallying; and (3) secure outsourced computation with input/output validation.

### 3.2. The Proposed Janus

Janus tackles the challenges of dynamic user participation, verifiability, and resistance to MIA that are not addressed in the state-of-the-art Flamingo (S&P'23). Specifically, it has the following three *key high-level technical ideas:*

(1) *Dual-server architecture and dynamic user participation.* Janus involves two servers, $S_0$ and $S_1$. Server $S_0$ is responsible for aggregating the masked updates, while server $S_1$ handles the aggregation of values associated with the commitments. The dual-server architecture prevents the servers from accessing the final aggregation results, thus effectively avoiding attacks such as model reversal and MIA, which are serious privacy leakage in a traditional single-server. Furthermore, there is no need to re-establish complex communication graphs when users join or leave. New users can participate in the new training process by simply generating their own public/private keys and obtaining the servers' public keys.

(2) *Lightweight components and efficient aggregation.*

Janus removes the requirement for users to share masking values with neighbors and avoids the costly peer-to-peer key negotiation required by the advanced Flamingo and BBSA. Instead, it applies OTP to mask the updates and encrypts the masking values through lightweight public key encryption. Then, each user only needs to send one message to $S_0$ and one to $S_1$, respectively. As a result, regardless of the number of users in the system, the computation overhead on each user remains constant.

(3) *Verifiability and privacy enhancement.* The separability of SHC enables users to locally validate the aggregated

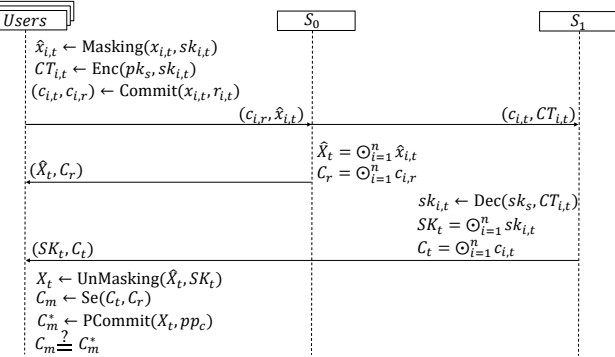

*Figure 1.* The Workflow of Janus. $S_0$ is the aggregation server and $S_1$ is the assistant server. It is worth noting that $S_0$ and $S_1$ do not need to interact throughout the process.

values, ensuring verifiability. In addition, its binding property prevents users from denying previously sent malicious messages when subsequent misbehavior is detected. This is a feature not available in other advanced schemes. Given the hiding of the SHC and the confidentiality of public key encryption, neither $S_0$ nor $S_1$ can access the received secret information. Combined with our dual-server architecture, higher security can be achieved.

Figure 1 shows the workflow of Janus. Subsequently, we provide a detailed description of Janus, noting that it is a generic construction. Thus, we assume the underlying public key encryption scheme is $\Pi_E = (\mathsf{Setup}, \mathsf{KeyGen}, \mathsf{Enc}, \mathsf{Dec})$, the OTP scheme is $\Pi_O = (\mathsf{Masking}, \mathsf{unMasking})$, and the SHC scheme is $\Pi_S = (\mathsf{Setup}, \mathsf{Commit}, \mathsf{Se}, \mathsf{PCommit}, \mathsf{Reveal})$, in which the setup parts of these schemes are completed in the **Setup** phase of Janus by default. Furthermore, Appendix B presents the tasks of each entity in different phases and provides an instantiation of the generic scheme to demonstrate its practicality. Specifically, Janus consists of the following four phases.

**(1) Setup.** The objective of this phase is to determine the public parameters $pp$ and specific cryptographic schemes, which ensure that subsequent schemes work properly. All participants are given the security parameter $\lambda$. Then the public parameters $pp$ are generated based on $\lambda$, including those used in the setup phase and public parameters generation in $\Pi_E, \Pi_O, \Pi_S$. Each user will generate their private keys $sk_{i,t}$ for the OTP. Server $S_1$ will generate its public/private keys $(pk_s, sk_s)$ and publish its public key to other participants. Subsequent communications between the users and the servers are encrypted with their respective public keys by default.

**(2) Masking and Report.** $U_{i,t}$ masks its local update $x_{i,t}$ through $\mathsf{Masking}(x_{i,t}, sk_{i,t})$ to get the masked update

*Table 2.* Comparison of Performance Analysis

| Scheme | Computation | | Communication | |
|---|---|---|---|---|
| | Client | Server | Client | Server |
| SecAgg (Bonawitz et al., 2017) | $\mathcal{O}(n^2 + md)$ | $\mathcal{O}(dn^2))$ | $\mathcal{O}(n + m)$ | $\mathcal{O}(n^2 + mn)$ |
| BBSA (Bell et al., 2020) | $\mathcal{O}(A^2 + lA)$ | $\mathcal{O}(n(A^2 + lA))$ | $\mathcal{O}(A^2 + l)$ | $\mathcal{O}(n(A^2 + l))$ |
| VeriFL (Guo et al., 2020) | $\mathcal{O}(n)$ | $\mathcal{O}(n + l)$ | $\mathcal{O}(n)$ | $\mathcal{O}(1) + \mathcal{O}(n)$ |
| ELSA (Rathee et al., 2023) | $\mathcal{O}(1 + l)$ | $\mathcal{O}(n + nl)$ | $\mathcal{O}(1)$ | $\mathcal{O}(n)$ |
| Flamingo (Ma et al., 2023) | R: $\mathcal{O}(L^2)$ 
 D: $\mathcal{O}(L^2 + \delta An + (1 - \delta)n + \epsilon n^2)$ | $\mathcal{O}(n + L^2)$ | R: $\mathcal{O}(l + A + L^2)$ 
 D: $\mathcal{O}(L^2 + L + \delta An + (1 - \delta)n)$ | $\mathcal{O}(L^3 + n(l + L + A))$ |
| Janus | $\mathcal{O}(1 + l)$ | $\mathcal{O}(n + nl)$ | $\mathcal{O}(1)$ | $\mathcal{O}(n)$ |

* Let $n, L, A$ denote the total number of clients, the number of decryptors, and the upper bound number of neighbors of a client, respectively, where $A = \log n$ in BBSA. $l$ denotes the dimension of the update. $\delta$ denotes the dropout rate, respectively. $\epsilon$ is the parameter of graph generation. R and D denote regular client and decryptor, respectively.

$\hat{x}_{i,t}$. Subsequently, $U_{i,t}$ encrypts the $sk_{i,t}$ using the public key of $S_1$ through $\mathsf{Enc}(pk_s, sk_{i,t})$ to get the ciphertext $CT_{i,t}$. To enable subsequent verifiability, $U_{i,t}$ generates a separable commitment to its local update $x_{i,t}$ using $\mathsf{Commit}(x_{i,t}, r_{i,t})$, where $r_{i,t}$ is a random blinder. The resulting commitment $c_{i,t}$ can be separated into two parts: $(c_{i,r}, c_{i,m})$, where $c_{i,r}$ is the commitment of blinder; and $c_{i,m}$ is the commitment of local update. Then $U_{i,t}$ sends $(\hat{x}_{i,t}, c_{i,r})$ to the aggregation server $S_0$ and $(c_{i,t}, CT_{i,t})$ to the assistant server $S_1$.

**(3) Collection and Aggregation.** In this phase, $S_0$ and $S_1$ will complete aggregation of users updates. Specifically, $S_0$ will aggregate the masked input updates from all users via $\hat{X}_t = \bigodot_{i=1}^{n} \hat{x}_{i,t} = \hat{x}_{1,t} \odot \hat{x}_{2,t} \odot ... \odot \hat{x}_{n,t}$. Then $S_0$ computes $C_r = \bigodot_{i=1}^{n} c_{i,r} = c_{1,r} \odot c_{2,r} \odot ... \odot c_{n,r}$. $S_0$ sends $(\hat{X}_t, C_r)$ to all users. $\hat{X}_t$ contains the updated aggregated values, and $C_r$ serves to validate the aggregated result in round $t$. $S_1$ first decrypts the ciphertext to get the $sk_{i,t}$ via $\mathsf{Dec}(sk_s, CT_{i,t})$. Then it can aggregate the $\bigodot_{i=1}^{n} sk_{i,t} = sk_{1,t} \odot sk_{2,t} \odot ... \odot sk_{n,t} = SK_t$. Furthermore, it computes the aggregated commitment value $\bigodot_{i=1}^{n} c_{i,t} = c_{1,t} \odot c_{2,t} \odot \cdots \odot c_{n,t} = C_t$, which enables subsequent users to verify the aggregation performed by $S_0$. Finally, $S_1$ sends $(SK_t, C_t)$ to all users.

**(4) UnMasking and Verification.** Users compute the final update and verify the aggregated result based on the values received from $S_0$ and $S_1$. Specifically, $U_{i,t}$ gets the final aggregation result through $X_t = \mathsf{UnMasking}(\hat{X}_t, SK_t)$, where the $X_t$ is the final aggregation result of the round $t$. To verify the correctness of the aggregation result, $U_{i,t}$ extracts the commitment value related to the updates through $C_m = \mathsf{Se}(C_t, C_r)$. $U_{i,t}$ then calculates the commitment value, which is only related to the updates through $C_m^* = \mathsf{PCommit}(X_t, pp_c)$, where the $pp_c$ is the public parameter of the underlying SHC. Finally, $U_{i,t}$ compares whether $C_m^*$ and $C_m$ are equal. If the two values are equal, the aggregated result is accepted as valid. Otherwise, it is rejected, and $U_{i,t}$ terminates the subsequent training.

## 4. Evaluation

### 4.1. Theoretical Analysis

Janus offers enhanced security compared to state-of-the-art schemes. As detailed in Appendix C, we provide a formal security analysis demonstrating that Janus is resistant to MIA and achieves multi-round security. A key advantage of Janus over schemes like Flamingo and BBSA lies in its ability to complete each aggregation round with fewer interactions. In these advanced schemes, additional communication with neighboring nodes is required to eliminate masks or perform decryption. By contrast, Janus significantly reduces interaction complexity. Furthermore, given that core operations such as commitments and encryptions are of constant complexity $\mathcal{O}(1)$, Janus achieves superior efficiency by maintaining a constant interaction count regardless of the number of users.

Notably, Janus is designed to be highly client-friendly, with system overhead independent of the number of users—a major limitation of previous schemes. Users only need two interactions with the servers before going offline, minimizing computation and communication demands. As a result, the aggregation process remains robust and accurate, even in the presence of user disconnections or dropouts. To highlight the efficiency of Janus, we focus on a single aggregation round. A detailed comparison of communication and computation overhead with other advanced schemes is presented in Table 2.

**Computation Cost.** The computation cost of each user consists of: (1) masking the local update via OTP; (2) encrypting the key of OTP by public key encryption; (3) committing the local update by SHC; (4) unmasking the global aggregation result; (5) separating message-only commitments from the full commitment; and (6) calculating the commitment value based on the unmasking result and comparing whether it is equal to the separated commitment value to complete the verification. All the above operations take only $\mathcal{O}(1)$. Therefore, the computation overhead of each user is constant. The computation cost of $S_0$ mainly consists of

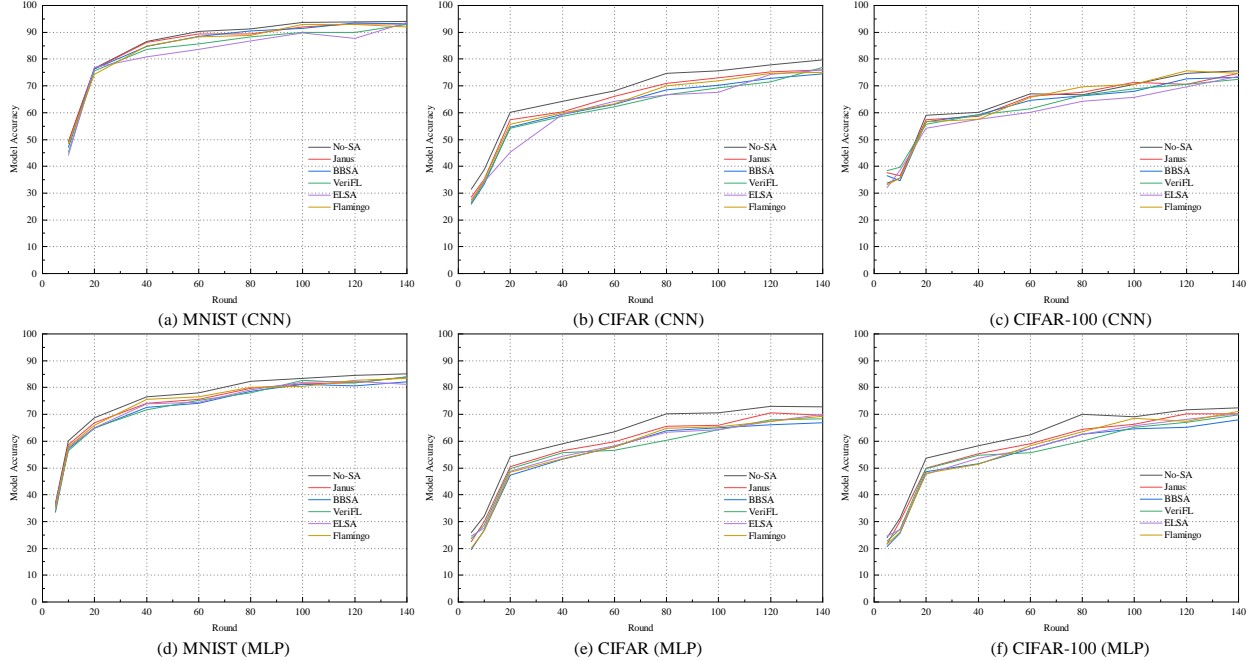

*Figure 2.* Test accuracy across different datasets and models.

aggregating the masking updates from users and the commitment of random numbers, which both take $\mathcal{O}(n)$. Thus, the total computation overhead grows linearly with the number of users. For $S_1$, the computation cost consists of: (1) decrypting the ciphertext of the private key of OTP; (2) aggregating the private keys for masking; and (3) aggregating the complete commitments for subsequent verification of the aggregation result from $S_0$. All above operations take $\mathcal{O}(n)$. Overall, the overhead of servers grows linearly with the number of clients, which takes $\mathcal{O}(n)$.

**Communication Cost.** Each user only needs to send one masked message to $S_0$ and one encrypted commitment to $S_1$. As a result, each user incurs a constant communication overhead. $S_0$ will send the aggregation result of the masking updates to all users, which takes $\mathcal{O}(n)$. $S_1$ sends both the aggregated OTP key and the full commitment to all users, which also incurs $\mathcal{O}(n)$ communication overhead. $S_1$ sends both the aggregated OTP key and the full commitment to all users, which also incurs $\mathcal{O}(n)$ communication overhead. Overall, the communication overhead on the server side scales linearly with respect to the number of users.

### 4.2. Experiments

To assess the practical performance of Janus, we carried out comprehensive experiments focusing on both effectiveness and efficiency. We further compared it with several representative advanced schemes. Our experimental setup includes a 13th Gen Intel(R) Core(TM) i7-13700KF 3.40 GHz processor with 32.0 GB of RAM, a 64-bit Windows 11 operating system, and an RTX 4070Ti GPU display adapter.

**Datasets and Models.** MNIST consists of 70,000 grayscale handwritten digit images (60,000 for training, 10,000 for testing), each 28x28 pixels. We simulated an environment with 100 users, each holding 600 local training samples. The global model for MNIST is a fully connected network with layers of size (784, 256, 10). CIFAR-10 includes 60,000 color images across 10 classes (50,000 training, 10,000 testing), using a CNN architecture with a batch size of 10, a learning rate of 0.001, and 100 training epochs. We employed SGD as the optimizer, with each user applying SGD once per global epoch.

**Baselines.** We implemented the original FL (No-SA), where the server aggregates plain updates in each training (McMahan et al., 2017). BBSA (Bell et al., 2020) optimizes the communication graph of the first mask-based SA scheme (Bonawitz et al., 2017). Meanwhile, Flamingo (Ma et al., 2023) introduces multi-round aggregation; VeriFL (Guo et al., 2020) achieves verifiability; and ELSA (Rathee et al., 2023) improves efficiency of the system. Additionally, Flamingo, VeriFL, and BBSA are designed to wait for messages from at least $t$ out of $n$ users. To handle user dropouts, we reconstruct the communication graph using only the responsive clients, resulting in better performance than the original approach.

**Model Performance.** To comprehensively evaluate the impact of the SA on the model training effectiveness, our

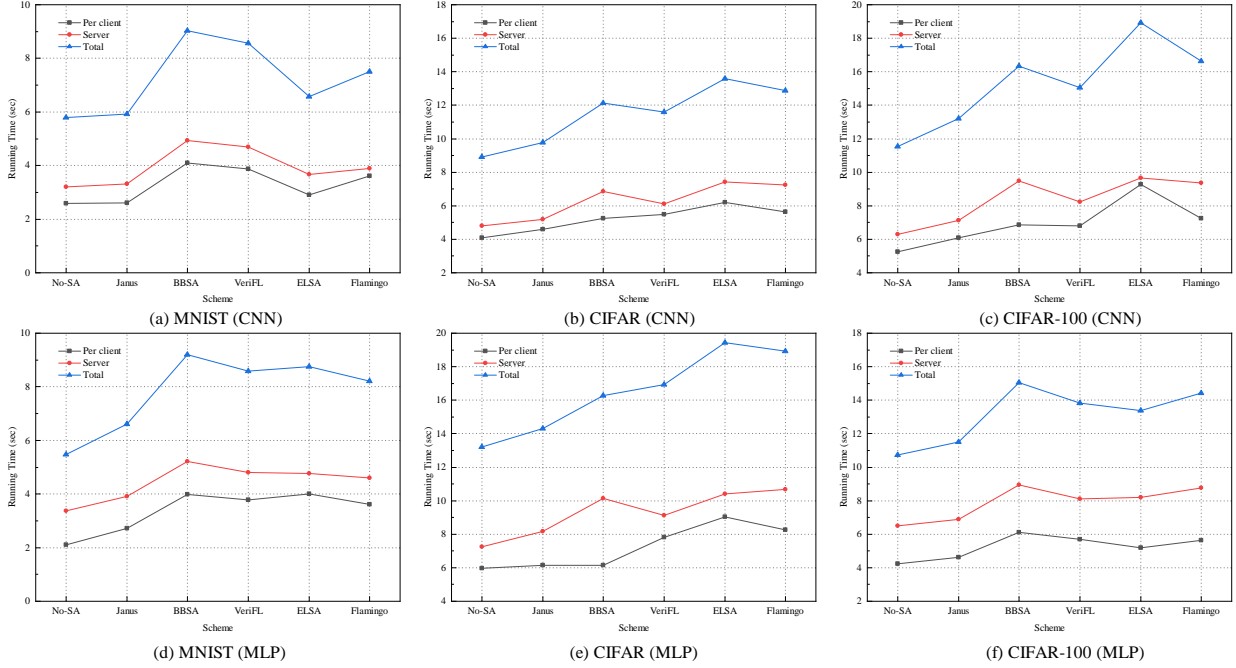

*Figure 3.* Computation overhead across different datasets and models.

experiments are carried out on different datasets and models. We conducted the training with 100 users and compared the test accuracy of our Janus with advanced schemes as shown in Figure 2. The experimental results lead to the following conclusions. The final test accuracy after model convergence differs only slightly between Janus and the compared schemes. The No-SA baseline achieves the highest accuracy, with Janus closely trailing. Notably, Janus is not limited to specific models or datasets. These results confirm its practicality and effectiveness in preserving accuracy under secure aggregation.

Specifically, on the MNIST dataset, No-SA achieves 94.1% test accuracy with CNN, while Janus reaches 93.18%. With MLP, No-SA reaches 85.04%, and Janus achieves 83.95%. Compared to other SA schemes, Janus maintains competitive accuracy. On the CIFAR dataset, No-SA achieves 77.8% with CNN and 72.8% with MLP, while Janus achieves 75.94% and 71.6%, respectively. Janus achieves test accuracy close to the No-SA baseline across different models and datasets.

**Computation Overhead.** Since masking-based schemes are vulnerable to user dropouts, we take this into account when implementing BBSA, VeriFL, and Flamingo. Specifically, we simulated a 10% user dropout rate. However, the waiting time to handle dropouts is typically much longer due to the complexity and variability of practical environments. Moreover, it is important to note that Flamingo inherently supports multi-round aggregation, while other

schemes lack this capability. To enable multi-round aggregation, we adapted them by repeatedly executing the single-round protocol to simulate multi-round aggregation. Although this approach is feasible, it introduces a considerable amount of additional computation overhead. This further highlights the advantages of our Janus, which is natively designed to support multi-round aggregation without incurring such overhead, thus demonstrating superior efficiency and scalability.

As shown in Figure 3, we compare the time overhead of different schemes, focusing on the completion time for a single aggregation round. Note that, given the structural differences between these schemes, the comparison includes only the stages that contribute significantly to the overall time consumption in each case. The computation overhead introduced by SA is within a practical and acceptable range, demonstrating its practicality in real-world applications. More importantly, Janus incurs significantly lower overhead, particularly on the user side, which enhances overall efficiency. This advantage stems from the use of lightweight cryptographic primitives that avoid costly operations such as secret sharing and Diffie-Hellman key exchange, thereby reducing the computation burden on users and contributing to its superior performance.

## 5. Conclusion

In this paper, we introduce a new cryptographic primitive, termed separable homomorphic commitment. Building on

this, we propose Janus, a generic dual-server multi-round secure aggregation scheme for federated learning. Specifically, Janus effectively tackles critical challenges, including dynamic user participation, verifiability, and resistance to model inconsistency attacks that are not considered in the advanced Flamingo. It significantly enhances security while simultaneously improving system efficiency, notably reducing per-user communication and computation overhead from logarithmic to constant scale with a tolerable impact on model accuracy. Both theoretical analysis and experimental evaluations consistently demonstrate its superior performance. Future work may focus on integrating Janus with advanced privacy-preserving techniques and designing secure mechanisms to detect and mitigate user-level poisoning attacks.

## Acknowledgements

The authors would like to thank the reviewers and area chair for their constructive feedback and support to improve this paper. This work was supported in part by the National Natural Science Foundation of China under Grant 62425205, Grant U21A20466, and Grant 62372108.

## Impact Statement

This paper presents work whose goal is to advance the field of privacy-preserving Federated Learning. There are many potential societal consequences of our work, none of which we feel must be specifically highlighted here.

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

# A. Related Works

The primary objective of federated learning is to preserve the privacy of local data while enabling its effective use in training the global model. This section reviews the works related to Janus.

**Masking-based SA.** Masking is a classic encryption based on OTP (Katz & Lindell, 2014). (Bonawitz et al., 2017) designed the first SA scheme (SecAgg) which using pairwise masks to hide individual inputs for FL. However, SecAgg involves a complete communication graph, which incurs heavy computation and communication overhead for each user linear in the number of participants. Subsequently, (Bell et al., 2020) replaced that with a $k$-regular graph of logarithmic degree, which greatly improved the efficiency while maintaining the security. (Stevens et al., 2022) replaced the standard mask with learning with errors mask and used verifiable secret sharing to prevent malicious users from distributing incorrect shares. (Sandholm et al., 2025) arranged the users in the system in the form of a ring chain, and the efficiency has been significantly improved. Additionally, the user dropout problem can be effectively solved. Most masking-based schemes require double masking in order to solve the problem of dropout. (Bonawitz et al., 2019) combined the random rotation technique to actively adjust the quantisation range of the model in order to reduce the model volume. To reduce the communication overhead, TurboAgg (So et al., 2021) divided $n$ users into $n/\log n$ groups and then used a multi-group loop structure for subsequent aggregation.

**MPC-base SA.** Multi-Party Computation (MPC) enables distrustful parties to jointly compute a target function while preserving privacy, which perfectly aligns with the SA. (Mohassel & Zhang, 2017) designed a scheme using secure two-party computation and proposed MPC-friendly alternatives to non-linear functions. Prio (Corrigan-Gibbs & Boneh, 2017) employed a novel technique known as SNIPs (Secret-shared Noninteractive Proofs), enabling servers to collaboratively verify a shared proof of correctness with minimal communication overhead. Prio+ (Addanki et al., 2022) replaced zero-knowledge proofs with Boolean secret sharing and share conversion protocols, boosting user-side performance. SAFELearn (Fereidooni et al., 2021) employed MPC to enable SA, resisting inference attacks with just two communication rounds while eliminating trusted third parties. (Gehlhar et al., 2023) proposed an MPC-based FL framework that combines SA with poisoning-resistant techniques, achieving privacy and robustness. (Ben-Itzhak et al., 2024) introduced the ScionFL, which efficiently handles quantized inputs while providing robustness against malicious clients and supporting various 1-bit quantization schemes.

**Attacks that Bypass SA.** Most existing schemes expose the aggregation results to both users and the server. However, this design introduces a potential vulnerability: a malicious server may exploit this visibility to circumvent the secure aggregation. (Pasquini et al., 2022) proposed model inconsistency attacks, where a malicious server can distribute different parameters to targeted and non-targeted users. This can trigger *dying-ReLU* and make the input of non-target users be zero. (So et al., 2023) noticed that when the trained model begins to converge, the user model changes little between one training step and the next. A malicious server can infer the updates of a user who participated in the previous round but did not participate in the subsequent round from the aggregation results. (Gao et al., 2021) proposed a scheme that can launch a category inference attack even in the presence of SA. To avoid this type of attack, when the users receive the model parameters, they need to verify whether the received parameters are consistent or not, and terminate the training if they are not. But this will increase the system overhead. (Fernández et al., 2021) applied differential privacy on the aggregated model to hide the aggregation results.

**Server-side Attacks and Defenses.** Membership inference attacks pose a potential threat from the server side in FL. Specifically, an adversary can determine whether some specific data records are part of the local training dataset of a target user only by accessing the model updates, either through a black-box or white-box approach. (Yeom et al., 2018) proposed the first label-based attack, which aims at predicting whether an instance is in the local data of the target user. The attacker leverages the target model's inferior performance on the test dataset to carry out the attack. (Chen & Vikalo, 2024) proposed a general method that allows the server to recover user training labels, applicable to various FL algorithms without assumptions on activation functions or batch label composition. (Shokri et al., 2017) designed an attack with partial output knowledge in a black-box scenario. Furthermore, (Salem et al., 2019) improved a new attack by using the maximum value of the model output confidence. (Zhuang et al., 2024) introduced layer substitution analysis, a novel technique for identifying layers that are critical for backdoor injection, making it well-suited for FL attacks. This technique enabled the development of two layer-wise backdoor attack strategies that implant backdoors into key layers and bypass advanced defenses without degrading main task accuracy.

Meanwhile, (Bonawitz et al., 2017) proposed the first SA scheme to compute the sum of model updates hiding personal information. Subsequently, a great deal of research has centred around SA. Techniques such as homomorphic encryption (Zhang et al., 2020a), differential privacy (Stevens et al., 2022), and multi-party computation (Bell et al., 2020) are used to construct

SA schemes to protect user privacy from attack by malicious servers. Cryptography-based SA aims to prevent attacks by hiding model updates from potential adversaries. Preserving the confidentiality of individual contributions, significantly reduces the risk of sensitive information leakage.

Recently, (Yueqi et al., 2024) identified a limitation in existing model poisoning attacks defenses: reliance on cross-client or global information, which leads to performance degradation or when there is a large number of malicious users. A crucial distinction between model poisoning attacks and benign model updates is then established by determining whether the update can be approximately reconstructed using distilled local knowledge. (Wu et al., 2024) proposed FedInverse, a framework designed to evaluate whether FL models are susceptible to model inversion attacks and quantify the associated data-leakage risks. (Garov et al., 2024) showed that all existing malicious server attacks can be identified through systematic checks. Furthermore, essential requirements for practical malicious server attacks were systematically established.

**Verifiability.** In addition, a malicious server may return incorrect aggregation results to gain unfair advantages or undermine system integrity. Such behavior poses serious security risks, as it can mislead users who rely on the aggregated outputs. Therefore, verifiable SA is necessary to ensure correct aggregation. (Zhang et al., 2020b) verified the aggregation result via homomorphic encryption and homomorphic hash function. Additionally, (Xu et al., 2019) verified masking-based SA using the same technique. (Guo et al., 2020) proposed a verification scheme that focuses on high-dimensional inputs. (Brunetta et al., 2021) proposed a non-interactive verifiable SA protocol, which requires users to create a tag for each input shares. In contrast, (Tsaloli et al., 2021) proposed a scheme that requires only a single tag for each user.

**Multi-round Setting and Dynamic Joining.** Model convergence in FL typically requires multiple rounds of training, with each round contributing incrementally to the overall performance of the global model. However, most existing state-of-the-art SA schemes are designed to support only a single round of aggregation. Beyond protecting user privacy in a single round of training, existing research has also explored privacy concerns that arise cumulatively over multiple rounds of training. (Nguyen et al., 2022) and (So et al., 2022) proposed two new schemes support asynchronous aggregation. (Guo et al., 2022) designed a multi-round SA protocol for reusable secrets, and it is mainly oriented towards scenarios with small inputs (the input vector with small values).

Recently, (Ma et al., 2023) proposed Flamingo, which has no restrictions on input value. (So et al., 2023) mitigated the privacy leakage involved in multi-round aggregation through client selection. Furthermore, the existing schemes do not support dynamic joining. Flamingo assumes that the set of all clients participating in the training is fixed before the training starts and some subset is selected from $n$ in each round $t$. Therefore, Flamingo does not support the user to dynamically add in the training process. Most existing schemes require reconstructing the communication graph upon user join or departure, and performing key negotiation with all other users, leading to significant communication and computation overhead. In addition, (Wang et al., 2023) explored cross-round aggregation of local models and proposed FedCDA, a novel method that constructs the global model by aggregating local updates from multiple rounds based on minimum divergence. To improve efficiency, FedCDA incorporates an approximation strategy to reduce the overhead of model selection.

## B. Detailed Janus and Its Instantiation

Figure 4 gives the full generic construction of Janus. Furthermore, an efficient instantiation of Janus is provided, where the underlying SHC is instantiated by Pedersen commitment, the public key encryption is realized via ElGamal, and the OTP employs normal addition encryption. Specifically, Janus consists of the following four phases.

*Setup.* This phase determines the public parameters of the system. Firstly, all participants agree on the security parameter $\lambda$. The public parameters of the cryptographic primitives are then generated based on $\lambda$. Define a tuple $(p, q, g, h)$, where $p$ is a randomly chosen prime of length $|q| = \lambda + \delta$, with $\delta$ being a specified constant; $q$ is a prime divisor of $p - 1$ such that $q = (p - 1)/\gamma$, where $\gamma$ is a specified small integer; and $g, h$ are random generators of $\mathbb{Z}_p^*$ with order $q$. $U_{i,t}$ generates its public/private keys $(pk_i, sk_i) = (g^{sk_i} \pmod{p}, sk_i)$, where the $sk_i \in \mathbb{Z}_p^*$. $S_1$ generates its public/private keys $(pk_s, sk_s) = (g^{sk_s} \pmod{p}, sk_s)$ where the $sk_s \in \mathbb{Z}_p^*$. Then $S_1$ and $U_{i,t}$ publish their public keys to all entities while keeping their private keys secretly.

*Masking and Report.* Each user $U_{i,t}$ trains local data $\mathcal{D}_{i,t}$ to get the update $x_{i,t}$ for round $t$. $U_{i,t}$ masks the updated vector by $\hat{x}_{i,t} = \mathsf{Masking}(x_{i,t}, sk_{i,t}) = x_{i,t} + sk_{i,t} \pmod{p}$, where $sk_{i,t} \in \mathbb{Z}_p^*$. Then $U_{i,t}$ encrypts the $sk_{i,t}$ via $CT_{i,t} = \mathsf{Enc}(pk_s, sk_{i,t}) = (g^{k_{i,t}} \pmod{p}, sk_{i,t} \cdot pk_s^{k_{i,t}} \pmod{p})$. Furthermore, $U_{i,t}$ commits the $x_{i,t}$ via $c_{i,t} = \mathsf{Commit}(x_{i,t}, r_{i,t}) = g^{x_{i,t}} h^{r_{i,t}} \pmod{p}$, where the $r_{i,t} \in \mathbb{Z}_p^*$ and $(c_{i,r}, c_{i,m}) = (h^{r_{i,t}} \pmod{p}, g^{x_{i,t}} \pmod{p})$.

1. **Setup.**

   – All parties get the security parameter $\lambda$.

   – This phase generates the public parameter $pp$ of the system, which contains the specific commitment, one-time pad, and public key encryption.

   – The assitant server $S_1$ generates its public/private keys $(pk_s, sk_s)$ and publishes its public key to all users.

   – Each user generates its public/private keys $(pk_i, sk_i)$ and publish its public key to servers $S_0$ and $S_1$. Subsequent user-server interactions are via public key encryption by default.

2. **Masking and Report.**

   – Each user computes $\hat{x}_{i,t} \leftarrow \mathsf{Masking}(sk_{i,t}, x_{i,t})$, where the $sk_{i,t}$ is the private key generated by user $U_{i,t}$ in round $t$.

   – Each user encrypts the private key in OTP $CT_{i,t} \leftarrow \mathsf{Enc}(pk_s, sk_{i,t})$, where the $pk_s$ is the public key of the assistant server $S_1$.

   – Each user generates the commitment $c_{i,t} = (c_{i,r}, c_{i,m}) \leftarrow \mathsf{Commit}(x_{i,t}, r_{i,t})$, where the $r_{i,t}$ is the blinder and $c_{i,r}, c_{i,m}$ is the commitment parts of blinder and message respectively.

   – Each user sends $(\hat{x}_{i,t}, c_{i,r})$ and $(CT_{i,t}, c_{i,t})$ to $S_0$ and $S_1$, respectively.

3. **Collection and Aggregation.**

   – $S_0$ collects the messages $(\hat{x}_{i,t}, c_{i,r})$ from users and parses as $x_{i,t}$ and $c_{i,r}$.

   – Then $S_0$ computes the $\bigodot_{i=1}^{n} \hat{x_{i,t}} = \hat{X}_t$ and $\bigodot_{i=1}^{n} c_{i,r} = C_r$.

   – $S_0$ sends the $\hat{X}_t$ and $C_r$ to all users.

   – $S_1$ collects the messages $(CT_{i,t}, c_{i,t})$ from users and parses as $CT_{i,t}$ and $c_{i,t}$.

   – $S_1$ decrypts the $sk_{i,t} \leftarrow \mathsf{Dec}(CT_{i,t}, sk_s)$ and computes $\bigodot_{i=1}^{n} sk_{i,t} = SK_t$.

   – $S_1$ computes the $\bigodot_{i=1}^{n} c_{i,t} = C_t$.

   – $S_1$ sends the $SK_t$ and $C_t$ to all users.

4. **UnMasking and Verification.**

   – Each user receives the message from $S_0$ and $S_1$, then it decrypts the ciphertext as $C_r$ and $\hat{X}_t$ using its private key $sk_i$.

   – Each user unmasks the aggregation via $X_t \leftarrow \mathsf{UnMasking}(SK_t, \hat{X}_t)$.

   – Each user computes the commitment about the input updates via $C_m \leftarrow \mathsf{Se}(C_t, C_r)$.

   – Each user generates the commitment via $C_m^* \leftarrow \mathsf{PCommit}(X_t, PP_c)$, which is related to the updates. $PP_c$ is the public parameter of the commitment scheme. Then $U_{i,t}$ compares $C_m^* \stackrel{?}{=} C_m$. If it is equal, then the aggregation result completed by $S_0$ is correct, otherwise, it is invalid. Once the aggregation results are found to be incorrect, users will terminate the subsequent training.

*Figure 4.* Detailed Construction of Janus.

Finally, $U_{i,t}$ sends $\{c_{i,r}, \hat{x}_{i,t}\}$ and $\{c_{i,t}, CT_{i,t}\}$ to $S_0$ and $S_1$, respectively.

*Collection and Aggregation.* Subsequently, $S_0$ receives the message from $U_{i,t}$. Then it computes $\bigodot_{i=1}^{n} \hat{x}_{i,t} = \hat{x}_{1,t} + \hat{x}_{2,t} + ... + \hat{x}_{n,t} = \hat{X}_t$ and $\bigodot_{i=1}^{n} c_{i,r} = h^{r_{1,t}} h^{r_{2,t}} ... h^{n,t} \pmod{p} = C_r$. Then $S_0$ sends $C_r$ and $\hat{X}_t$ to all users. When the $S_1$ receives the message from $U_{i,t}$. It first decrypts $sk_{i,t} = \mathsf{Dec}(sk_s, CT_{i,t}) = sk_{i,t} \cdot pk_s^{k_{i,t}} (g^{k_{i,t}^{sk_s}})^{-1} \pmod{p}$. Subsequently, it computes $\bigodot_{i=1}^{n} c_{i,t} = c_{1,t} c_{2,t} ... c_{n,t} \pmod{p} = C_t$. Then it computes $\bigodot_{i=1}^{n} sk_{i,t} = sk_{1,t} + sk_{2,t} + ... + sk_{n,t} = SK_t$. Finally, $S_1$ sends the $C_t$ and $SK_t$ to all users.

*Unmasking and Verfication.* When $U_{i,t}$ receives the message from $S_0$ and $S_1$. Firstly, $U_{i,t}$ computes the aggregation result via $X_t = \mathsf{Unmasking}(\hat{X}_t, SK_t) = \hat{X}_t - SK_t$. To verify the validity of the aggregation results, $U_{i,t}$ separates the parts

of the commitments that are only relevant to the input updates by $\mathsf{Se}(C_t, C_r) = C_m$. Then $U_{i,t}$ makes a commitment to the aggregation result from $S_0$ via $\mathsf{PCommit}(X_t, pp_c) = g^{\hat{X}_t} \pmod{p} = C_m^*$, where $pp_c$ is the public parameters of the underlying SHC. Eventually, $U_{i,t}$ checks whether $C_m^* \stackrel{?}{=} C_m$ holds. If so, it indicates that the aggregated result $\hat{X}_t$ from $S_0$ is correct; otherwise, the result is deemed invalid. $U_{i,t}$ will refuse to accept the results of the aggregation and abort the subsequent training.

**Correctness.** The correctness of this instantiation is guaranteed as long as all entities follow the protocol honestly, ensuring that each user obtains both the correct aggregation result and valid verification. It is not hard to prove this due to the correctness of the underlying public key encryption, OTP, and SHC. Specifically, we assume that the aggregation server $S_0$ receives all masked-input and performs Janus correctly, the following condition holds.

$$
\begin{aligned}
\bigodot_{i=1}^{n} \hat{x}_{i,t} &= \hat{x}_{1,t} + \hat{x}_{2,t} + ... + \hat{x}_{n,t} \\
&= x_{1,t} + sk_{i,t} + x_{2,t} + sk_{2,t} + ... + x_{n,t} + sk_{n,t} \\
&= \bigodot_{i=1}^{n} x_{i,t} + \bigodot_{i=1}^{n} sk_{i,t} \\
&= X_t + SK_t,
\end{aligned} \tag{8}
$$

where the $\bigodot_{i=1}^{n} sk_{i,t}$ is computed by $S_1$. The final aggregation result is $\bigodot_{i=1}^{n} x_{i,t} = \bigodot_{i=1}^{n} \hat{x}_{i,t} - \bigodot_{i=1}^{n} sk_{i,t} = X_t$. If the validation passes, the following condition holds.

$$
\begin{aligned}
C_t &= g^{x_{1,t}} h^{r_{1,t}} g^{x_{2,t}} h^{r_{2,t}} ... g^{x_{n,t}} h^{r_{n,t}} \\
&= g^{x_{1,t}+x_{2,t}+...+x_{n,t}} h^{r_{1,t}+r_{2,t}+...+r_{n,t}}, \\
C_r &= h^{r_{1,t}+r_{2,t}+...+r_{n,t}}, \\
C_m &= C_t/C_r = g^{x_{1,t}+x_{2,t}+...+x_{n,t}}, \\
C_m^* &= g^{X_t}.
\end{aligned} \tag{9}
$$

If the aggregation result $X_t$ from $S_0$ is correct, then the $C_m^* = C_m$ always holds.

## C. Security Analysis

In this section, we intend to demonstrate the security of our generic construction. We first give the threat model and prove that our Janus can protect the privacy of users' local updates and the aggregated updates. Finally, we give the security proof of single-round and multi-round respectively.

### C.1. Threat Model

All users agree to publish the final results of model aggregation only to each user, but not to the servers, in order to resist MIA. These users share a common interest in both soundness—ensuring they receive the correct global model aggregation updates from untrusted servers—and the confidentiality of local model updates from each other and the servers.

The specific assumptions in our paper are as follows: $S_0$ and $S_1$ will not collude but may perform incorrect aggregation. Janus also allows for up to $n-2$ clients to collude. Specifically, even if the servers aggregate incorrect results, Janus provides verifiability, which enables us to detect such behavior and mitigate the associated risks.

If the server colludes with up to $n-2$ clients, it can only obtain the additive result of the remaining two uncolluding clients. This result is an aggregation of two encrypted or obfuscated values, making it impossible to recover each uncolluding user's specific gradient information. This ensures that the colluding entities cannot initiate an MIA or access the private information of the remaining two non-colluding clients. When $n-2$ clients collude, this assumption is even weaker, as the absence of server involvement further limits the accessible information, making it even harder to extract useful data.

If only a single server is corrupted, this does not compromise individual user privacy. For instance, with server $S_0$, as long as the underlying encryption algorithm is secure, the server cannot access the user-submitted private data without the user's private key. Similarly, for server $S_1$, the security of the underlying SHC ensures that its hiding properties prevent $S_1$ from

obtaining any private information. In conclusion, the assumptions of our scheme are reasonable and well-supported. In addition, we assume the channel between each user and the servers is secure, which allows each entity to authenticate the incoming messages and prevent outsiders from injecting their responses. Furthermore, we assume that there is no collusion between all entities in the system. Our security proofs are based on this threat model.

### C.2. Privacy from Users

In the "honest but curious" setting, each user will honestly adhere to the protocol but attempt to infer the local gradients of clients and the aggregated gradients. Therefore, we can use the standard simulation proof for multi-party computation protocols to demonstrate the privacy of our generic construction. We first consider privacy protection against honest-but-curious clients who hold their own local gradients and have access to the global gradients. Specifically, let $\Pi$ denote the proposed Janus involving $n$ users $C_1, C_2, ..., C_n$ and two servers $S_0$ and $S_1$. Each user holds a local update gradient $x_i$, and Janus securely computes the aggregated global update $X$. All participants may attempt to infer additional information, the $\Pi$ satisfies the following privacy guarantee:

- For each honest-but-curious user $C_i$, the user learns nothing beyond its own local gradient $x_i$ and the final global aggregated gradient $X$. Formally, for each $C_i$, there exists a $\mathcal{PPT}$ simulator $\mathcal{S}_i$ such that:

$$\{\mathbf{View}_\Pi(C_i)\} \approx \{\mathcal{S}_i(x_i, X)\}, \tag{10}$$

  where $\mathbf{View}_\Pi(C_i)$ denotes the view of $C_i$ during the real execution of $\Pi$, $x_i$ is the $C_i$'s local updates and $X$ is the final global aggregated result.

- For $S_0$ and $S_1$, they learn nothing beyond the masked aggregated results and the aggregated results of masks. This can ensure they will learn nothing about the final global aggregated gradient $X$, thus resisting the MIA. Formally, for $S_0$ and $S_1$, there exists a $\mathcal{PPT}$ simulator $\mathcal{S}_{server}$ such that:

$$\{\mathbf{View}_\Pi(S_0, S_1)\} \approx \{\mathcal{S}_{server}(\hat{X}, CT)\}, \tag{11}$$

  where $\mathbf{View}_\Pi(S_0, S_1)$ denotes the view of two servers during the real execution of $\Pi$, $\hat{X}$ is the masked aggregation result, and $CT$ is the ciphertext of masks.

Given any subset $\mathcal{U} \subseteq \mathcal{C}$ of the users, where the $\mathcal{C}$ is the set of all users in the system ($|\mathcal{C}| = m$). Let the $\mathbf{REAL}_{\mathcal{U}}^{\mathcal{C}, \lambda}(\{(\hat{x}_{i,t})\}_{i \in \mathcal{C}}, (c_{1,r}, c_{2,r}...c_{m,r}))$ be a random variable representing the ioint view of the users in $\mathcal{U}$. This suggests that all these honest but curious clients learned the aggregation of the gradients of all clients and their own gradients.

---

**Functionality $\mathcal{F}_s$**

Parties: users $1, \ldots, N$ from $\mathcal{S}_t$ and two servers $S_0$ and $S_1$.
Parameters: corrupted rate $\eta$, number of participating training clients per-round n.

- $\mathcal{F}_s$ receives a set of corrupted parties $\mathcal{C}$ from the adversary $\mathcal{A}$, where the $|\mathcal{C}|/|\mathcal{S}_t| \leq \eta$.

- For each round $t$:
  1. $\mathcal{F}_s$ receives a set of $N$ clients $\mathcal{S}_t$ and updates $x_{i,t}$ from clients $i \in \{\mathcal{S}_t \setminus \mathcal{C}\}$.
  2. $\mathcal{F}_s$ sends $\mathcal{S}_t$ to $\mathcal{A}$ and requests a set $M_t$. $\mathcal{F}_s$ computes the $X_t = \bigodot_{i \in \{M_t \setminus \mathcal{C}\}} x_{i,t}$ if $M_t \subseteq \mathcal{S}_t$ and continues; otherwise $\mathcal{F}_s$ sends abort to all honest participants.
  3. There are two scenarios based on whether the servers are corrupted by $\mathcal{A}$ as follows.
     - Corrupted: $\mathcal{F}_s$ outputs $X_t$ to all the participants corrupted by $\mathcal{A}$.
     - Not corrupted: $\mathcal{F}_s$ requests a mask $SK_t$ from $\mathcal{A}$ and outputs $X_t \odot SK_t$ to $S_0$.

---

*Figure 5.* Ideal functionality for Janus.

### C.3. Single-Round Security

**Theorem C.1. (Security of Janus)** *Let the security parameter be $\lambda$ and $n$ be the number of users for aggregation in each round. Assuming the existence of secure OTP, SHC, and public key encryption. Our generic construction can securely*

*realize the ideal functionality $\mathcal{F}_s$ under the presence of a static adversary controlling $\eta$ fraction of $n$ users (and the server $S_1$) as shown in Figure 5.*

$$\mathbf{REAL}_{\Pi,\mathcal{A}}^{\mathcal{F}_s,\mathcal{F}_{sum}^t}(\lambda, n, x_{\mathcal{S}_t}) \approx \mathbf{IDEAL}_{\mathcal{F}_s,\mathcal{S}}^{\mathcal{F}_{sum}^t}(\lambda, n, x_{M_t}). \tag{12}$$

**Proof.** We first prove the security of a single-round aggregation. Our generic scheme (denoted as $\Pi$) securely realizes the ideal functionality $\mathcal{F}_{sum}^t$ (Figure 6) in the random oracle model. We can find from the ideal function $\mathcal{F}_{sum}^t$ that it is the $M_t$ sent by the adversary $\mathcal{A}$ that determines the actual result. We assume the $\mathcal{A}$ controls a set of clients and denote the set of corrupted clients as $\mathcal{C}$.

**Event 1.** We start with the servers not being corrupted by the $\mathcal{A}$. Now, we first build a simulator $\mathcal{S}$ in the ideal world, running $\mathcal{A}$ as a subroutine. Specifically, the simulation for round $t$ is as follows.

1. $\mathcal{S}$ receives a set $M_t$ from the adversary $\mathcal{A}$.

2. $\mathcal{S}$ acquires $Z_t$ from the $\mathcal{F}_{sum}^t$.

3. *Masking and Report.* $\mathcal{S}$ interacts with $\mathcal{A}$ as in the masking and report phase and acts as honest users in $i \in \{M_t \setminus \mathcal{C}\}$ with the masked updates $x_{i,t}'$ such that the $Z_t = \bigodot_{i \in \{M_t \setminus \mathcal{C}\}} x_{i,t}'$. Here the input update $x_{i,t}'$ and the mask $sk_{i,t}$ are generated by $\mathcal{S}$.

4. *Collection and Aggregation.* In this phase, $\mathcal{S}$ interacts with $\mathcal{A}$, where $\mathcal{A}$ performs as an honest participant in the collection and aggregation of $\Pi$.

5. *UnMasking and Verfication.* $\mathcal{S}$ interacts with $\mathcal{A}$ as honest participants in the unmasking and verfication phase.

6. In the above steps, if all honest participants would abort in the protocol in this round of aggregation, then $\mathcal{S}$ sends abort to $\mathcal{F}_{sum}^t$. Finally, $\mathcal{A}$ outputs the value at random and terminates this aggregation.

---

**Functionality $\mathcal{F}_{sum}^t$**

Parties: users from $\mathcal{S}_t$ and two servers.
Parameters: corrupted rate $\eta$.

- $\mathcal{F}_{sum}^t$ receives a set of corrupted participants $\mathcal{C}$ from the adversary $\mathcal{A}$ and $x_{i,t}$ from clients $i \in \{\mathcal{S}_t \setminus \mathcal{C}\}$.

- $\mathcal{F}_s$ sends $\mathcal{S}_t$ to $\mathcal{A}$ and requests a set $M_t$. $\mathcal{F}_s$ computes the $Z_t = \bigodot_{i \in \{M_t \setminus \mathcal{C}\}} x_{i,t}$ if $M_t \subseteq \mathcal{S}_t$ and continues; otherwise $\mathcal{F}_s$ sends abort to the all honest participants.

- For each round $t$:
    1. $\mathcal{F}_s$ receives a set of $N$ clients $\mathcal{S}_t$ and updates $x_{i,t}$ from clients $i \in \{\mathcal{S}_t \setminus \mathcal{C}\}$.
    2. $\mathcal{F}_s$ sends $\mathcal{S}_t$ to $\mathcal{A}$ and requests a set $M_t$. $\mathcal{F}_s$ computes the $Z_t = \bigodot_{i \in \{M_t \setminus \mathcal{C}\}} x_{i,t}$ if $M_t \subseteq \mathcal{S}_t$ and continues; otherwise $\mathcal{F}_s$ sends abort to the all honest participants.
    3. There are two scenarios based on whether the servers are corrupted by $\mathcal{A}$ as follows.
        - Corrupted: $\mathcal{F}_s$ outputs $X_t$ to all the participants corrupted by $\mathcal{A}$.
        - Not corrupted: $\mathcal{F}_s$ requests a mask $SK_t$ from $\mathcal{A}$ and outputs $Z_t \bigodot SK_t$ to $S_0$.

---

*Figure 6.* Ideal functionality for Report and Collection in round $t$.

We construct a series of hybrid execution programs from the real world to the ideal world.

**Hybrid 1.** The view of $\mathcal{A}$ in the real-world execution is the same as the ideal world, when $\mathcal{S}$ has actual inputs from honest participants $\{x_{i,t}\}, i \in \mathcal{S}_t \setminus \mathcal{C}$ including the individual masks $sk_{i,t}$ and the $SK_t$.

**Hybrid 2.** $\mathcal{S}$ does not use the actual masks in OTP between honest participants. It generates a random mask $sk_{i,t}'$ from the $\{0,1\}^\lambda$, then it computes the corresponding OTP ciphertext as $\hat{x}_{i,t}'$. We argue the view of $\mathcal{A}$ in this hybrid is computationally indistinguishable from the previous hybrid 1 as follows.

Firstly, the mask $sk_{i,t}$ is computed from the space $\mathcal{R}_C$ of the OTP, and the mask $sk_{i,t}'$ is randomly sampled in the ideal world. Let the $M_t$ denote the set of users chosen by $\mathcal{A}$ in the ideal world. $\mathcal{A}$ in the ideal and real world can observes

Masking$(x_{i,t}, sk_{i,t})$ between a user $i \notin M_t$ and a user $i \in M_t$. This indistinguishability stems from the selection of random masks in the specific underlying OTP. Secondly, $\mathcal{A}$ can observe the ciphertexts generated from the $sk'_{i,t}$. The distribution of the ciphertexts is computationally indistinguishable from what $\mathcal{A}$ observes in the real world, assuming the security of the underlying OTP.

**Hybrid 3.** In this hybrid, instead of using OTP with actual personal mask $sk_{i,t}$ randomly selected from the space $\mathcal{R}_C$, $\mathcal{S}$ uses masks randomly sampled from $\{0,1\}^\lambda$. Before the proof, we model the generation of masks as a random oracle $\mathcal{O}_R$ (see more details in the prior work (Bonawitz et al., 2017)). For $\forall i \in \{M_t \setminus \mathcal{C}\}$, the $\mathcal{S}$ samples $sk'_{i,t}$ randomly and programs $\mathcal{O}_R$ as $sk'_{i,t} = \hat{x}_{i,t} \oslash x_{i,t}$, where the $\hat{x}_{i,t}$ is observed in the real world and the $\oslash$ denotes the inverse operation of $\odot$. From the perspective of $\mathcal{A}$, the distributions of $\hat{x}_{i,t}$ in this hybrid and the previous one are statistically indistinguishable.

Additionally, $\mathcal{A}$ learns the $sk'_{i,t}$ in the clear for $i \in M_t$ in the real and ideal world. The distributions of $sk'_{i,t}$ are identical. However, $\mathcal{A}$ learns nothing about $sk'_{i,t}$ for $i \notin M_t$ in both worlds because of the semantic security of the underlying OTP. From the view of $\mathcal{A}$, this hybrid is computationally indistinguishable from the previous hybrid.

**Hybrid 4.** In this hybrid, instead of controlling the random oracle as in the previous hybrid, $\mathcal{S}$ will program the random oracle $\mathcal{O}_R$ as $sk'_{i,t} = \hat{x} \oslash x'_{i,t}$. Specifically, the $x'_{i,t}$s are chosen such that $\bigodot_{i \in \{M_t \setminus \mathcal{C}\}} x_{i,t} = \bigodot_{i \in \{M_t \setminus \mathcal{C}\}} x'_{i,t}$. From the view of $\mathcal{A}$, this hybrid is the same as the previous one, which can be derived from Lemma 6.1 of the prior work (Bonawitz et al., 2017).

**Hybrid 5.** Similar to the previous operation, this hybrid replaces the mask of honest participants with the result from the random oracle. $\mathcal{S}$ will abort if the $\mathcal{A}$ would cheat by sending invalid masked updates to $\mathcal{S}$. In the phase of unmasking and verification, the $\mathcal{A}$ would cheat by sending different $M_t$ to $\mathcal{F}^t_{sum}$. $\mathcal{S}$ will simulate the following protocol (see as Lemma C.2) and output whatever the protocol outputs. It is identical to the previous hybrid by doing this.

The final hybrid precisely represents the execution of the ideal world. The aforementioned events indicate that our system is secure in the ideal world with a single round process.

**Event 2.** In this event, the server is not corrupted by $\mathcal{A}$, the whole simulation is the same as Event 1, except that the $\mathcal{S}$ will program the masks added by the $\mathcal{A}$ in each step.

We complete the proof that for any single round $t$, the protocol $\Pi$ always securely realizes the ideal functionality $\mathcal{F}^t_{sum}$ in the presence of a static malicious adversary.

**Lemma C.2.** *Assume there exists a PKI and a secure signature scheme, there are $3\zeta$ participants with at most $\zeta$ colluding malicious participants. Specifically, each party has an input bit of $0$ or $1$ from a server. There exists a one-round protocol enabling each honest participant to determine whether the server sent the same value to all honest participants.*

**Proof.** When an honest participant receives at least $2\zeta$ messages with the same value, it indicates that the server has sent the same value to all honest participants. In the given system, the threshold of $2\zeta$ identical messages can only be met if a large majority of honest participants have received the same value. Specifically, let the total number of participants in the system be $n = t_h + t_m$, where $t_h$ denotes the number of honest participants and $t_m$ denotes the number of malicious participants. For security and consistency in distributed protocols, the parameter $\zeta$ is set such that $t_h > 2\zeta$. When an honest participant receives no fewer than $2\zeta$ identical messages, it can confidently be concluded that at least $\zeta + 1$ of these messages were sent by distinct honest participants, ensuring consistency of the message content. Hence, it can be inferred that the server has broadcast the same value to all honest participants.

Conversely, if an honest participant receives fewer than $2\zeta$ messages with the same value, this suggests that the server may have sent different messages to different participants during the communication process. Since the number of identical messages received by honest participants falls short of forming a consensus of $2\zeta$, it implies that the server may have engaged in malicious behavior by sending inconsistent messages to various honest participants. To ensure the security and consistency of the protocol, the honest participant will abort the protocol execution. This abort mechanism effectively prevents potential security threats and data integrity issues that could arise due to inconsistent messages from the server.

### C.4. Multi-round Security

Our threat model assumes the corrupted rate is $\eta$, which means that $\mathcal{A}$ controls $\eta \cdot n$ clients throughout the total $T$ rounds. In order to prove the security of the multi-round scheme on the basis of the above single-round security proof. The mask $sk_{i,t}$ computed from $\mathcal{O}_R$ of the underlying SHC $\Pi_\mathcal{C}$. Let $\Delta_t$ denote the distribution of the adversary's view $\mathcal{A}$ in the single

round $t$, and let $T$ be the total number of rounds required for model convergence. If there exists an adversary $\mathcal{B}$, and two rounds of aggregation $t_1, t_2 \in [0, T]$, where $\mathcal{B}$ can distinguish between $\Delta_{t_1}$ and $\Delta_{t_2}$, then we can construct an adversary $\mathcal{A}$ breaks the security of the underlying $\Pi_\mathcal{C}$. Let the challenger in the security game of $\Pi_\mathcal{C}$ as $\mathcal{S}$. Specifically, there exists two worlds ($b = 0$ or $1$) for the $\mathcal{O}_R$ game. $\mathcal{S}$ uses a random function when $b = 0$. When $b = 1$, $\mathcal{S}$ executes the actual protocol $\Pi_\mathcal{C}$. Then we build the $\mathcal{A}$ as follows. On input $t_1, t_2$ from $\mathcal{B}$, the $\mathcal{A}$ asks for $sk_{i,t}$ for all honest participants in the round $t_1$ and $t_2$. $\mathcal{A}$ can compute the masked updates from the $\Pi$ prescribed. It generates two views $\Delta_{t_1}, \Delta_{t_2}$ and sends them to the $\mathcal{B}$. Finally, $\mathcal{A}$ outputs whatever $\mathcal{B}$ outputs as the answer.

## C.5. Resisting MIA

The MIA is effective primarily because the server is aware of the final aggregated result. If the server can manipulate the parameters sent to different clients, it can introduce inconsistencies that influence the model training process. The key to resisting this attack is to ensure that all clients start with the same initial model parameters. This uniformity can be achieved through two main approaches: using a public bulletin board where the initial parameters are posted for everyone to see, or through mutual agreement among clients to verify that the parameters they receive are indeed consistent across the network. The public bulletin board approach suffers from centralized dependency, information leakage, and scalability issues, while the mutual agreement method has high communication complexity, scalability limitations, and is vulnerable to Sybil attacks. Both methods face challenges in maintaining consistency and security as the number of clients increases.

A significant advancement introduced by Janus is its novel design where the aggregation results are made visible only to the users. In Janus, each user computes the final result locally, rather than relying on the server. As a result, even if the server $S_0$ distributes inconsistent model parameters to different users, it remains unaware of the actual aggregated model. This paradigm shift ensures that the servers cannot gain insight into the final result, thus preventing them from launching MIA.

Additionally, we assume that $S_0$ and the users will not collude. In other words, the server cannot conspire with any users to manipulate the aggregation process. By decentralizing the aggregation computation and keeping the final result private among the users, Janus effectively mitigates the risk of a successful MIA. This approach not only enhances the security of the federated learning framework but also reinforces the privacy and trustworthiness of the system by limiting the server's influence over the final model.

