# OpenReview forum: "Janus: Dual-Server Multi-Round Secure Aggregation with Verifiability for Federated Learning"
_ICML.cc/2025/Conference — ICML 2025 poster_

### Official Review · Reviewer_nXLy · 2025-03-10

**Overall Recommendation:** 2

**Summary:**

The paper proposes a new secure aggregation protocol for federated learning. This protocol relies on two non-colluding servers: one to aggregate masked gradients and the other aggregating one-time-pad masks. Compared to other similar schemes, the masks here are not secret-shared with a graph of neighbours, thereby enabling client dropouts etc. Also, due to the integration of a new cryptographic primitive (separable homomorphic commitments), the protocol provides security against model inconsistency attacks. The authors also implement and compare their protocol to prior works, confirming training accuracy and performance improvements.

## update after rebuttal
Thanks for participating in the rebuttal process. After re-evaluating the paper while considering the rebuttal, there was unfortunately no increase in the score. While I'm confident that the paper will improve significantly by implementing some of the proposed changes, we can only evaluate what was submitted. In particular, the definition of the separation operation is not sufficiently clear, and the main body of the paper does not demonstrate that Pedersen commitments satisfy the required homomorphism property. Additionally, some arguments provided in the rebuttal are not sound; for example, a simple (hardware-accelerated) hash verification can also be performed in O(l) and should be less costly than using exponentiation-based commitment schemes.

**Claims And Evidence:**

Claims regarding the efficiency and accuracy of the proposed secure aggregation scheme are substantiated with experiments.

**Essential References Not Discussed:**

In the introduction the authors claim that currently most secure aggregation schemes rely on a double masking approach. However, this ignores an entire class of secure aggregation schemes based on MPC / distributed aggregators. Likewise, such works are not discussed in Appendix A. Examples include:
- Fereidooni et al.: "SAFELearn: Secure Aggregation for private FEderated Learning"
- Gehlhar et al.: "SafeFL: MPC-friendly framework for Private and Robust Federated Learning"
- Ben-Itzhak et al.: "ScionFL: Efficient and Robust Secure Quantized Aggregation"

**Experimental Designs Or Analyses:**

The experimental design for verifying performance overhead and accuracy seem appropriate, covering asymptotic and empirical results.

**Methods And Evaluation Criteria:**

The presentation of the new cryptographic primitive is not convincing. First of all, some parts of the definition are not clear, e.g., if Commit gives a tuple (c_m, c_r), then Se is a trivial tuple access rather than requiring any computation. Moreover, there is no proof demonstrating that the implemented instantiation (sketched in Appendix B) indeed fulfils the security properties (the security proof in Appendix C only covers the generic construction).

**Other Comments Or Suggestions:**

The paper is fairly well written and mostly easy to follow. The proposed protocol has clear benefits over prior works, combining desirable features with reduced performance overhead. However, there are also several negative aspects as discussed in this review, preventing to recommend acceptance.

With respect to the potential vulnerabilities wrt MIA attacks, it is unclear why other simple solutions like publishing a signed hash of the aggregated model on a widely-visible medium or bullet-in board are not considered for users to verify that they have not been targeted.

The use of the notion "semi-trusted" is not clear. In §1, the authors state that semi-trusted servers could deliberately mishandle some gradients. It should be clarified if "semi-trusted" = "semi-honest" as typically semi-honest servers are assumed to not deviate from the protocol.

Section 4.1 claims that Janus offers enhanced security over prior works. However, it should be noted that if the non-collusion assumption does not hold, there are significantly more severe consequences as the attacker then can single out individual gradients whereas this would still not be possible when relying on secret-shared masks.

Minor/typos:
- Line 221: Pdeersen

**Other Strengths And Weaknesses:**

Strengths

+ New cryptographic primitive that might be of independent interest
+ Reduced overhead compared to related works (both asymptotically and concretely)
+ Support for client drop outs and security against MIA attacks

Weaknesses

- Definition of new cryptographic primitive not clear
- Only sketch for possible instantiation of new primitive
- No security proofs that the sketched instantiation fulfils all properties

**Questions For Authors:**

- Do you have a security proof for your instantiation of the separable homomorphic committment scheme?

**Relation To Broader Scientific Literature:**

The paper tries to enhance efficiency and accuracy of secure aggregation schemes for federated learning, while also providing robustness against model inconsistency attacks.

**Theoretical Claims:**

Claims on the security of the proposed instantiation of the SHC scheme are not substantiated with a proof.

---

> ### Author Rebuttal · Authors · 2025-03-31
>
> Dear Reviewer nXLy,
>
> Thank you for your valuable feedback and suggestions. Below, we address each of your concerns.
>
> **1. Definition of SHC (W1&Q1):**
>
> The output of the Commit algorithm is c, but c is not a simple tuple. In fact, we intend to convey that the complete commitment can be split into c_m and c_r via computation, rather than directly storing. We will revise the wording. The SHC is an abstract blueprint. In existing secure commitment schemes, any scheme that meets the properties we have outlined qualifies as a SHC. In fact, we instantiate SHC as the Pedersen commitment. Furthermore, when discussing the properties of SHC (lines 215-234), we have already discussed that the Pedersen commitment possesses the properties of SHC. Thus, its security is inherently guaranteed by the security of the underlying commitment, eliminating the need for a separate proof.
>
> **2. References:**
>
> Masking-based and MPC-based SA are orthogonal, with the former relying on mask generation and elimination, and the latter on distributed computations. To offer a more comprehensive view of SA, we will include the following excellent MPC-based works, including the references you recommended in the next version.
>
> **MPC-base SA.** Multi-Party Computation (MPC) enables distrustful parties to jointly compute a target function while preserving privacy, which perfectly aligns with the SA. (Mohassel & Zhang, 2017) designed a scheme using secure two-party computation (2PC) and proposed MPC-friendly alternatives to non-linear functions. Prio (Corrigan-Gibbs & Boneh, 2017) employs a novel technique known as SNIPs (Secret-shared Noninteractive Proofs), enabling servers to collaboratively verify a shared proof of correctness with minimal communication overhead. Prio+(Addanki et al., 2022) replaces zero-knowledge proofs with Boolean secret sharing and share conversion protocols, boosting client-side performance. SAFELearn (Fereidooni et al., 2021) employs MPC to enable SA, resisting inference attacks with just two communication rounds while eliminating trusted third parties and supporting client dropouts. (Gehlhar et al., 2023) proposed a MPC-based FL framework that combines SA with poisoning-resistant techniques, achieving privacy and robustness. (Ben-Itzhak et al., 2024) introduced a novel scheme (ScionFL) that efficiently handles quantized inputs while providing robustness against malicious clients and supporting various 1-bit quantization schemes.
>
> **3. Misunderstanding of Instantiation (W2):**
>
> We have already provided a full instantiation in lines 640-714, not just the new primitive. The component primitives can be replaced with any instantiation that satisfies the conditions, enabling compatibility with different systems.
>
> **4. Security of Instantiation (W3):**
>
> We have already proven the security of the generic construction in lines 742-957. The generic construction is an abstract structure, and there are multiple possible implementations. Proving the security of the generic construction ensures the security of all instantiated schemes. Meanwhile, some existing generic constructions [1-3], ensure security in the same way we do. In summary, the instantiation based on the generic construction provide equivalent security guarantees.
>
> [1] Nguyen et al. Multimodal private signatures, CRYPTO22.
>
> [2] Luo et al. Generic construction of trace-and-revoke inner product functional encryption, ESORICS22.
>
> [3] Yuen et al. DualRing: generic construction of ring signatures with efficient instantiations, CRYPTO21.
>
> **5. Other Defense of MIA (S1):**
>
> The method you mentioned defends against MIA by adding an extra operation required in each round, while FL typically requires multiple rounds to converge. In contrast, our Janus naturally provides this defense upon completing aggregation, without any additional steps. Second, verifying the signed hash in every round could impose a significant operational burden on clients. We believe this effectively addresses your concerns.
>
> **6. Synonym and Typos (S2):**
>
> The term 'semi-honest' is more suitable for our situation, and we will correct it.
>
> **7. Misunderstanding of Assumption (S3):**
>
> Compared to schemes that rely on the non-collusion assumption, our scheme enhances security by resisting MIA attacks and enabling verifiable results. Furthermore, it offers greater functionality by supporting dynamic participation and multi-round aggregation. Additionally, our assumption aligns with that of prominent works such as references [1, 2], both of which also rely on the non-collusion assumption, making it a common and reasonable choice in this field.
>
> [1] Ma et al. Flamingo: Multi-round single-server secure aggregation with applications to private federated learning.
>
> [2] Rathee et al. Elsa: Secure aggregation for federated learning with malicious actors.
>
> Thank you for your time and effort. We will update the content in the next version. If you have any further concerns, please feel free to let us know.

---

### Official Review · Reviewer_LvMf · 2025-03-11

**Overall Recommendation:** 4

**Summary:**

This paper aims to address the challenges in existing secure aggregation (SA) schemes, including scalability with dynamic user participation, vulnerability to model inconsistency attacks (MIA), and the lack of verifiability in server-side aggregation results. The motivation is compelling and beneficial to the advancement of federated learning (FL). The proposed approach introduces a dual-server SA architecture and a new cryptographic primitive, namely Separable Homomorphic Commitment (SHC), which together enable key properties such as scalability, verifiability, and MIA security.

**Claims And Evidence:**

The detailed design sufficiently supports the authors’ claims. First, the dual-server architecture eliminates the need for heavy communication graphs, thereby efficiently addressing the challenge of dynamic user participation. Moreover, the integration of Separable Homomorphic Commitment further mitigates potential attacks from malicious servers, including privacy leakage via MIA and incorrect aggregation behavior.

**Essential References Not Discussed:**

The current related works provide sufficient insight into the recent progress in SA research.

**Experimental Designs Or Analyses:**

The experimental designs are sound and valid. The authors consider both dataset selection and multi-party scenario simulation. The chosen datasets are appropriate for covering diverse scenarios and models, particularly since the proposed approach is a generic construction not limited to a specific dataset or model.

The only concern is that the authors could provide further discussion on the rationale behind setting the user dropout rate at 10%, even though the theoretical analysis has demonstrated that the scheme can tolerate up to n−2 colluding users.

**Methods And Evaluation Criteria:**

The authors provide both theoretical and experimental evaluations. The theoretical analysis demonstrates the advantages of their proposal over existing SOTA schemes in terms of computation and communication costs. The experimental analysis further validates the feasibility of their approach in the FL setting. Specifically, the authors employ two datasets, MNIST and CIFAR, to assess the impact of their proposal. They evaluate training effectiveness and communication time, comparing their method with both the original FL framework and SOTA SA schemes. The final results demonstrate that the proposed approach is practical and effective in preserving model accuracy while enhancing security.

**Other Comments Or Suggestions:**

1. A proof sketch could be included in the main body to provide a brief understanding of the security proofs, even for readers who do not consult the appendix.
2. The potential applications of the SHC primitive could be further discussed to broaden its impact.

**Other Strengths And Weaknesses:**

Strengths:
- The design of dual-server-based SA protocol is concise yet effective, which successfully addresses the remaining challenges faced by existing SA protocols.
- The proposed SHC is valuable for designing SA protocols in the FL setting and holds independent theoretical interest in commitment research.
- The clearly defined threat model and solid security proofs not only substantiate their claims but also inspire further research on SA in the dual-server setting.

Weakness:
- The authors did not discuss the rationale behind setting the user dropout rate at 10% in Section 4.2.

**Questions For Authors:**

1. What is the rationale behind setting the user dropout rate at 10% in the experimental section?
2. Are there other potential applications for the newly proposed SHC primitive?

**Relation To Broader Scientific Literature:**

Like existing literature, this work focuses on achieving secure aggregation (SA) in the FL setting. In addition to maintaining high accuracy and privacy protection, as ensured by existing SA methods, this work further achieves scalability, verifiability, and MIA security. Notably, its key components, including the previously mentioned dual-server architecture and SHC primitive, are lightweight yet capable of effectively guaranteeing the targeted properties.

**Theoretical Claims:**

The threat model and security proofs are reasonable and correct. Specifically, the newly defined threat model within the proposed dual-server architecture is sound. The proofs concerning privacy, single-round security, multi-round security, and resistance to MIA are correct  and provide solid support for the authors' claims. Additionally, the authors provide clear examples of collusion resistance.

One possible suggestion is that the authors could briefly include a proof sketch in the main body, even though the detailed proofs are provided in the appendix.

---

> ### Author Rebuttal · Authors · 2025-03-31
>
> Dear Reviewer LvMf,
>
> We sincerely appreciate your valuable feedback. Below we provide a point-by-point response.
>
> **1. Dropout rate (W1&Q1):**
>
> To ensure a fair comparison, it is crucial to recognize that mask-based approaches (including BBSA, VeriFL, and Flamingo in Section 4.2) fundamentally require solving the client dropout to achieve correct aggregation. While secret sharing can address dropouts, its reconstruction overhead grows substantially with higher dropout rates (as it must recover masks for all dropped clients). In contrast, our Janus framework maintains normal aggregation regardless of dropout events, giving it inherent scalability advantages. Existing evaluations of BBSA/Flamingo already consider dropout rates up to 30%, whereas our experiments focus on demonstrating Janus’s superiority under dropout conditions. Even at the modest 10% dropout rate we tested, Janus shows significant performance gains—advantages that would only amplify with increasing dropout rates. Thus, selecting a 10% dropout rate is both methodologically sound (aligning with prior work’s evaluation scope) and sufficient to highlight our core contribution: dropout-robust aggregation. We will explicitly discuss this rationale in our revision.
>
> **2. More Applications (Q2):**
>
> We appreciate this insightful question regarding SHC's broader applicability. It is crucial to note that SHC is fundamentally a general-purpose cryptographic primitive, not limited to the specific applications we discussed. While our paper focuses on its application to SA in federated learning, SHC's architectural flexibility makes it adaptable to any scenario requiring both confidentiality and verifiability, particularly in distributed systems with accountability requirements. Specifically, SHC's cryptographic separation of concerns makes it uniquely suitable for: (1) medical data federation where SHC enables privacy-preserving auditing, (2) dual-server e-voting systems needing verifiable tallying, and (3) secure outsourced computation requiring input/output validation. We will add a discussion in the next version.
>
> Thank you again for your insightful comments. We will integrate these clarifications in the next version. Please don't hesitate to share any further concerns.

---

### Official Review · Reviewer_uPUn · 2025-03-13

**Overall Recommendation:** 4

**Summary:**

This paper proposes Janus, a secure aggregation scheme based on dual servers for FL, whose core innovation lies in breaking through the communication constraints of the traditional single-server architecture: through the design of a bidirectional interaction protocol that supports multiple rounds of aggregation and verifiable results. Janus takes the lead in realizing the enhancement of users' offline freedom under the collaborative verification mechanism of the two servers. The dual servers each have their own role to constrain each other and securely solve the aggregation problem in FL. For dynamic user scenarios, Janus lifts the strong dependency of users online, so that new users can join or leave without reconfiguring the communication topology, which significantly improves system scalability. To support the architecture, the paper innovatively proposes SHC, which provides a cryptographic foundation for the dual-server paradigm. The experimental part verifies the theoretical analysis through side-by-side comparison of multiple datasets and similar advanced schemes, which improves the security while Jauns has good efficiency. Finally, the security analysis confirms the advantage of the scheme in balancing between efficiency and security.

**Claims And Evidence:**

Yes.

**Essential References Not Discussed:**

Essential related works for understanding the key contributions of the paper have been appropriately cited and discussed.

**Ethical Review Concerns:**

Affirmed.

**Experimental Designs Or Analyses:**

Yes, I checked these designs and found no issues.

**Methods And Evaluation Criteria:**

Yes.

**Other Comments Or Suggestions:**

Suggestion: The potential of combining Janus with differential privacy or homomorphic encryption is not discussed, limiting its application to higher privacy demanding scenarios.

**Other Strengths And Weaknesses:**

Strengths:

1.	Innovative architectural design. I like the overall solution idea of this paper, where the dual servers each have their own role to constrain each other and securely solve the aggregation problem in federated learning.

2.	Efficiency and scalability. Reducing client communication and computation overhead from logarithmic level to constant level significantly improves the efficiency.

3.	New technologies and applications. The perfect fit of SHC and dual-server architecture can provide new ideas of privacy assurance for subsequent dual-server systems.

4.	Enhanced security. Protecting user gradient privacy while supporting verifiable aggregation results, effectively resisting model inconsistency attacks. The defense analysis of MIA could provide new ideas for such research.

Weaknesses:

1.	Insufficient details of dynamic engagement. While dynamic user joining is supported, it is not clearly stated how to handle dynamic updates of user keys (e.g., key rotation mechanism), which may introduce the risk of long-term key compromise.

2.	Not reader friendly enough. Although the security assumptions in this paper are to some extent reasonable and feasible, the main paper is not friendly to read, and the paper only portrays the threat model in the appendices. However, I believe that highlighting these in the main paper can eliminate concerns about the security assumptions.

**Questions For Authors:**

1.	Despite the numerous capabilities and benefits of the proposed scheme, what limitations does it currently have?
2.	This paper focuses on multi-round secure aggregation. What challenges does existing research face in achieving multi-round secure aggregation?

**Relation To Broader Scientific Literature:**

This paper realizes multi-round secure aggregation, which supports dynamic user updates while verifying the aggregation results, in addition to effectively resisting the model inconsistency attack, which are security issues that have not been simultaneously addressed in previous studies. This paper provides useful ideas for subsequent research.

**Theoretical Claims:**

Yes, I checked the correctness and proof of security of the scheme in this article and both are correct.

---

> ### Author Rebuttal · Authors · 2025-03-31
>
> Dear Reviewer uPUn,
>
> Thank you for your valuable comments. We address your concerns as follows.
>
> **1. Dynamic Engagement (W1):**
>
> Our scheme enables dynamic participation where clients can join or leave at any time without compromising security. The process is designed as follows: new clients simply (1) obtain system public parameters, (2) generate their key pairs, and (3) complete one training round with the initial model to receive the global update. Departing clients gracefully exit by ceasing submissions, while key updates follow the same streamlined joining procedure. Importantly, all participant changes occur without requiring reconstruction of the communication graph, maintaining both security and system efficiency.
>
> **2. Writing (W2):**
>
> The current version outlines our security assumptions in the abstract and Lines 88-108, with complete formal specifications provided in Appendix C.1. We will incorporate a more detailed discussion of these assumptions in the next version.
>
> **3. Future Work (	Q1):**
>
> Building on our current work, we will explore methods to resist client poisoning attacks, a common challenge faced by existing masking and encryption-based schemes. Additionally, we will focus on further enhancing both the efficiency and security of our approach.
>
> **4. Challenges of Multi-round (Q2):**
>
> The challenges of multi-round aggregation are already discussed in Lines 42 to 98 of the paper. Specifically, the core challenges regarding this aspect are threefold: (1) most existing schemes require regenerating system parameters for multi-round aggregation, i.e., repeating a single round aggregation many times to realize multiple rounds; (2) practical training scenarios with dynamic participation (users joining/leaving) typically demand complex communication graph reconstruction; and (3) while existing approaches need dedicated, time-consuming operations to achieve both multi-round execution and MIA resistance, our scheme inherently resists MIA during SA without additional overhead.
>
> We greatly appreciate your feedback and will ensure these clarifications will be included in the next version. If you have any further concerns, please let us know.

---

### Official Review · Reviewer_e8s4 · 2025-03-13

**Overall Recommendation:** 1

**Summary:**

Janus proposes a 2-server aggregation protocol in which one of the servers provides the aggregated masked results and the other the aggregated masks and aggregated commitments so that clients can check that the results are consistent. The protocol also prevents the server from ever learning the output to improve the privacy guarantee. The paper goes on to explain the results of some experiments that check that ML still works if the gradient is aggregated using various different means of implementing discrete addition.

**Claims And Evidence:**

All the real justificaiton for the relevant claims is in the supplementary material so I haven't really read it closely. The protocol is fairly simple so it isn't too hard to understand what it does anyway.

**Essential References Not Discussed:**

There are no missing references that stand out.

**Experimental Designs Or Analyses:**

The experiments seem pointless so I wouldn't know what to check.

**Methods And Evaluation Criteria:**

I don't understand the point of the experiments section. Doesn't Janus just compute the exact (albeit discretized) aggregation of the contributions from clients, just like every other protocol for secure aggregation. If so how could the results of the ML experiments be any different and what is added by providing them.

**Other Comments Or Suggestions:**

In your definition of SHC your c_m is never used as an input to any of the functions, so there is no implicit requirement on it at all. I think I know what you want because I assume you just want Pedersen commitments and to do the obvious thing at each point, but if you are going to specify SHC as a new primitive at least define it precisely. It isn't really a new primitive if the reader has to just guess what you mean from their knowledge of how (the rather old primitive of) Pedersen commitments work.

**Other Strengths And Weaknesses:**

There are a few problems with this paper.

The assumption that the server might be malicious to the point of running MIA attacks but not be capable of corrupting even a single client in a dynamic client system seems very unrealistic. Thus the value of the server "not finding out the output" seems pretty low.

Aggregation is easy in the two server model. That the inputs can be additively secret shared is decades old folklore and that one of them can be expanded from a key to reduce communication is also decades old folklore.

The secret key aggregation seems wrong, you seem to be assuming that the keys can be aggregated and expanded to give the same thing as if they have been expanded and then aggregated. Something like this is possible with RLWE c.f. ACORN or WILLOW, but you don't mention anything about that.

If you meant to aggregate the entire masks at S_1, this seems to give the server S_1 the opportunity to add any amount to the final aggregation completelynegating the point of the commitments (the only part of the protocol that isn't old folklore). Also this would substantially increase the communication overhead destroying probably all of your communicaiton advantage over other recent protocols.

You claim the protocol is multi-round and dynamic but say nothing about how a new joining client could get access to the up to date model. Obviously as the server isn't suposed to learn this it should be secret and you would need some plan to deal with it.

**Questions For Authors:**

What situation is the no corrupt client assumption reasonable in?

What do you mean by the secret key aggregation?

How do clients joining learn the model?

What are the experiments intended to check, that isn't obvious from the fact you are replacing addition with something that is bit for bit the same under normal conditions?

Why does the functionality you define for your security proofs end with S_1 receiving the output when they are just a support server and hte output seems like it goes to the clients in the protocol? Is it because you don't know how to prove S_1 can't mess up the result that the clients receive? (this somewhat negates the point of the commitments and the reason you can't prove that is impossible is because there is an attack)

**Relation To Broader Scientific Literature:**

The key contributions seem pretty minimal, I don't think they contribute to the discussion usefully.

**Theoretical Claims:**

I didn't check the proofs.

---

> ### Author Rebuttal · Authors · 2025-03-31
>
> Dear Reviewer e8s4,
>
> We sincerely appreciate your time and effort spent on our work. We have noticed that most of your concerns stem from misunderstandings about our paper. Below, we will first clarify these misunderstandings point by point.
>
> **Misunderstanding 1. Security assumptions (W1&Q1):**
>
> We clarify that we do not assume the server cannot corrupt a single client—rather, it ensures security even if the server corrupts up to n-2 clients. For large-scale FL (n ≫ 1), corrupting n-2 clients is impractical, making this a realistic assumption (lines 88-101 and Section C.1). Additionally, our assumption aligns with the prominent works like references [1] and [2]. Such an assumption represents a widely accepted and theoretically sound approach within this research domain.
>
> [1] Ma et al. Flamingo: Multi-round single-server secure aggregation with applications to private federated learning.
>
> [2] Rathee et al. Elsa: Secure aggregation for federated learning with malicious actors.
>
> **Misunderstanding 2. Unrelated Techniques and Secret Key Aggregation (W2,3&Q2):**
>
> Our work does not involve secret sharing and key expansion as you mentioned. Additionally, our work goes beyond merely implementing aggregation. We have also introduced multi-round, verifiable aggregation that is resistant to MIA, among other enhancements. Finally, secret key aggregation works as follows: clients first encrypt masks using S1's public key, then S1 aggregates, and finally users verify correctness by cross-checking with S0's aggregated values. This mutual verification between S0 and S1 provides the security guarantee without time-consuming RLWE, et al. The details are in Lines 264-317, with formal correctness analysis in Lines 715-740.
>
> **Misunderstanding 3. Security Proof (Q5):**
>
> Our security proof is correct: our scheme (lines 252-317) is consistent with the ideal function (lines 799-812, 846-862), where the server holds the masked aggregated gradients (Lines 288-317), and the client holds the gradient aggregation values. The security proof methodology follows the state-of-the-art scheme Flamingo (SP23). The only identified issue is a minor typo in the function definition (S1 should be S0), which does not introduce the inconsistency you raised. This typo will be corrected in the next version.
>
> Next, we will address your concerns point by point.
>
> **Concern 1. Malicious S_1 (W4):**
>
> Our scheme is already designed to resist the type of attack you mentioned. Specifically, S_0 and S_1 constrain each other, and users can validate the aggregated results of the two servers with the SHC. If either server attempts manipulation, clients detect inconsistencies during validation and abort participation. After theoretical analysis (Table 2), this does not increase our communication overhead.
>
> **Concern 2. New User Join (W5&Q3):**
>
> The process for a new user to obtain the model is simple and straightforward. Specifically, the user only needs to retrieve the aggregation information from S0 and S1, and then, like any other user, they can locally obtain the model parameters. Moreover, the new user can not only acquire the model parameters but also verify whether the parameters provided by the two servers are correct. This is detailed in Lines 304 to 317 of the paper.
>
> **Concern 3. Experiment (Q4):**
>
> Our experiments aim to evaluate the trade-offs introduced by SA (increased training time, potential efficiency overhead) while demonstrating that these costs remain acceptable compared to the security and function benefits. Our experimental design follows the relevant literature such as [1,2] in the same field. Therefore, our experimental design is reasonable and well validates the theoretical analyses in Section 4.1.
>
> [1] Wang et al. VOSA: Verifiable and oblivious secure aggregation for privacy-preserving federated learning.
>
> [2] Ma et al. Flamingo: Multi-round single-server secure aggregation with applications to private federated learning.
>
> We greatly appreciate your feedback and will incorporate these clarifications in the next version. Thank you again, and we look forward to your response.

---

### Decision · Program_Chairs · 2025-05-01

**Decision:**

Accept (poster)

**Comment:**

This paper introduces Janus, a secure aggregation (SA) protocol for multi-round federated learning utilizing a dual-server architecture. Janus makes several valuable contributions by addressing key limitations in existing SA schemes. Notably, it provides an efficient mechanism for handling client dropouts across multiple rounds, a significant practical challenge. Furthermore, it incorporates client-side verifiability, allowing clients to confirm the correctness of the server's aggregation, enhancing trust and robustness. Supportive reviewers highlighted these features, along with the convincing security analysis against n-2 colluding clients and compelling experimental results demonstrating significant efficiency gains over the recent Flamingo protocol. While concerns were raised regarding the novelty and the practicality of the non-colluding dual-server assumption, these should be considered in context. The non-collusion assumption is standard for dual-server models and enables the enhanced security/verifiability features Janus offers. Regarding novelty, while individual components like secret sharing and commitments are known, their specific integration within Janus to achieve efficient multi-round SA with robust dropout handling and client-side verifiability appears to be a novel and worthwhile contribution, potentially offering advantages over prior work as argued by the authors. Criticisms regarding the MIA robustness mechanism and limited baselines, while technically valid points for discussion, do not negate the paper's core strengths. The client-side computation demonstrably mitigates certain MIA vectors compared to server-centric approaches, and the strong performance against the relevant Flamingo baseline supports the claims of efficiency. Overall, Janus presents a well-motivated and technically sound approach that advances the state-of-the-art in practical secure aggregation by effectively balancing efficiency, dropout tolerance, and verifiability, making it a valuable contribution to the field.